**The influence of water table depth on evapotranspiration in the Amazon arc of deforestation**
Authors: John O'Connor[1], Maria J. Santos[2], Karin T. Rebel[1] and Stefan C. Dekker[1,3]
[1]Copernicus Institute of Sustainable Development, Department Environmental Sciences, Utrecht University, The
Netherlands
[2]University Research Priority Program in Global Change and Biodiversity and Department of Geography,
University of Zürich, Switzerland
[3]Faculty of Management, Science and Technology, Open University, Heerlen, The Netherlands
*Correspondence to*: John C. O Connor  (j.c.oconnor@uu.nl)
Abstract:
The Amazon rainforest evapotranspiration (ET) flux provides climate regulating and moisture provisioning
ecosystem services through a moisture recycling system. The dense complex canopy and deep root system
creates an optimum structure to provide large ET fluxes to the atmosphere forming the source for precipitation.
Extensive land use and land cover change (LULCC) from forest to agriculture in the arc of deforestation breaks
this moisture recycling system. Crops such as soybean are planted in large homogeneous monocultures and the
maximum rooting depth of these crops is far shallower than forest. This difference in rooting depth is key as
forests can access deep soil moisture and show no signs of water stress during the dry season while in contrast
crops are highly seasonal with a growing season dependant on rainfall. As access to soil moisture is a limiting
factor in vegetation growth, we hypothesised that if crops could access soil moisture they would undergo less
water stress and therefore would have higher evapotranspiration rates than crops which could not access soil
moisture.
We combined remote sensing data with modelled groundwater table depth (WTD) to assess whether vegetation
in areas with a shallow WTD had higher ET than vegetation in deep WTD areas. We randomly selected areas of
forest, savanna and crop with deep and shallow WTD and examined whether they differ on MODIS
Evapotranspiration (ET), Land Surface Temperature (LST) and Enhanced Vegetation Index (EVI), from 2001 to
2012, annually and during transition periods between the wet and dry season. As expected, we found no
differences in ET, LST, and EVI for forest vegetation between deep and shallow WTD, which because of their
deep roots could access water and maintain evapotranspiration for moisture recycling during the entire year. We
found significantly higher ET and lower LST in shallow WTD crop areas than in deep WTD during the dry
season transition, suggesting that crops in deep WTD undergo higher water stress than crops in shallow WTD
areas.
The differences found between crop in deep and shallow WTD, however, are of low significance with regards
the moisture recycling system as the difference resulting from conversion of forest to crop has an overwhelming
influence (ET in forest is $\approx$ 2 mm day$^{-1}$ higher than that in crops) and has the strongest impact on energy balance
and ET. However, access to water during the transition between wet and dry seasons may positively influence
growing season length in crop areas.

## 1 Introduction

The Amazon rainforest has been reduced to 80% of its original size due to deforestation over the past few decades (Davidson et al., 2012). Land use and land cover change (LULCC) from forest and savanna to agricultural land disrupts the Amazonian water cycle due to changes in evapotranspiration, infiltration, and runoff (Fearnside, 1997; Lawrence and Vandecar, 2014). Changes in evapotranspiration result in major changes to the water energy balance, as forest vegetation has high evapotranspiration rates and is replaced with agricultural vegetation with lower evapotranspiration which results in a lower latent heat flux and higher sensible heat flux (Swann et al., 2015). In addition, a decline in evapotranspiration reduces the available atmospheric moisture which can reduce rainfall. Differences in vegetation structure are suggested to be the main drivers affecting the evapotranspiration rates. Three major land cover classes can be identified at the Amazon arc of deforestation; forest, savanna (Brazilian cerrado, here we use savanna to keep terms equal with the land cover classification used) and agriculture. Forest vegetation has the highest total leaf surface area while savanna has a lower leaf surface area owing to its mixed structure of grasses shrubs and trees with a more open canopy and agricultural vegetation usually has a lower leaf area (Asner et al., 2003; Costa et al., 2007). This difference in leaf area lowers the potential surface area for both interception evaporation and transpiration. In addition, the rooting depth of forest savanna and agricultural vegetation differs greatly (Costa and Foley, 2000). Forest vegetation have deep roots which facilitate access to deep soil moisture maintaining their supply of water necessary for photosynthesis even during the dry season. Therefore, forest evapotranspiration remains high throughout the year, unaffected by periods of low rainfall (Maeda et al., 2017; Staal et al., 2018). While the rooting depth of savanna tree species have been shown to be deep, the savanna landscape also contains more open shallow rooted shrubs and grasses. Following LULCC from forest or savanna to agriculture the new vegetation cover lacks deep roots and therefore no longer accesses deeper soil moisture. Over the past few decades, the developing agricultural industry driven by international demand encouraged extensive LULCC (Brando et al., 2014; Foley et al., 2007; Sampaio et al., 2007) concentrated along the southern and eastern edge of the Amazon in an area known as the arc of deforestation (Costa and Pires, 2010; Malhi et al., 2008). LULCC negatively impacts the ecosystem service provision of the Amazon including highly valuable services such as carbon storage and sequestration and moisture recycling and regulation. However, little is known whether LULCC that occurred in areas with a shallow WTD facilitates access to water and leads to higher vegetation productivity and evapotranspiration compared to areas with a deep WTD. Understanding the effect that LULCC has on evapotranspiration is important as the loss of evapotranspiration impacts both climate and precipitation on local and regional scales.

Local climate can be impacted by LULCC due to changes in the energy balance as loss of evapotranspiration reduces latent heat and increases sensible heat. Studies in the Amazon have shown that temperatures increase on average 1.4 °C with a max of 7 °C following conversion to crop (Badger and Dirmeyer, 2015). The seasonal impact of LULCC is particularly strong during the dry season as crop evapotranspiration is at its lowest, latent heat flux can be reduced by 78% and the sensible heat flux can increase by 85% relative to forest (Ponte De Souza et al., 2011). The loss of evapotranspiration impacts rainfall both locally and on the continental scale. Evapotranspiration returns water to the atmosphere where it can precipitate again either in situ or be carried further downwind (Eltahir and Bras, 1994). Large forests like the Amazon, because of their density and extent create large evapotranspiration fluxes, leading to underpressure over land and the pressure differences draw moisture towards land (Makarieva

and Gorshkov, 2007; Sheil, 2014). As high as 70% of rainfall in the Amazon and southern Brazil is a result of
Amazonian evapotranspiration (van der Ent and Savenije, 2011). This evapotranspiration precipitation cycle is
highly important in both maintaining the forest itself but also providing precipitation to non forested areas. LULCC
reduces the evapotranspiration and breaks this moisture recycling system resulting in lower rainfall locally and
downwind. The seasonal loss of evapotranspiration in crop areas during the dry season is of great significance,
evidence already suggests that LULCC has resulted in a lengthening of the dry season (Costa and Pires, 2010;
Debortoli et al., 2017). Model simulations predict that if deforestation continues by 2050 the loss of
evapotranspiration will result in a negative effect further reducing forest cover and evapotranspiration (Foley et
al., 2007; Spracklen et al., 2012). The conversion of forest and savanna to agricultural land in Brazil is driven by
an increasing demand for agricultural production which has almost doubled since 2000 (Zalles et al., 2019);
however, losses in evapotranspiration could lead to subsequent losses in agricultural productivity as rainfall is
reduced and the growing season is shortened (Oliveira et al., 2013).

Crops in the Amazon arc of deforestation are predominantly rainfed and as such impacted by the high seasonality
in rainfall unseen in forest vegetation. Forest vegetation provides an optimum structure for evapotranspiration due
to its tall complex, dense canopy and deep root systems which can access deep soil moisture stores and maintain
high transpiration rates even during periods of low rainfall (Nepstad et al., 1994; Sheil, 2014). Savanna has a mixed
composition, with both trees and grass layers, more open canopy and lower leaf area. Savanna trees can have a
deep rooting depth (> 10 m) facilitating access to deep soil water (Canadell et al., 1996). Agricultural crops are
known to contribute much less to evapotranspiration as a result of their shorter canopy and simpler structure
(Fearnside, 1997). In addition, agricultural crops lack the deep root systems of forest which are credited for
maintaining evapotranspiration throughout the dry season (Nepstad et al., 1994). In theory, if vegetation continues
to access the water table within the root zone then this vegetation will continue to transpire during periods of
reduced rainfall. Thus limited access to soil moisture is an important limiting factor for photosynthesis and
transpiration. Shallow water table depths across South America are widely distributed and correspond to an area
of approximately 36% of the Amazon (Fan and Miguez-Macho, 2010). We hypothesize that areas of shallow water
table depth (WTD) allow shallow rooted vegetation to access soil moisture, potentially facilitating vegetation
productivity and higher evapotranspiration when compared to areas of deep WTD. Experimental manipulation of
WTD using sub irrigation systems of soybean demonstrated that shallow WTD benefitted productivity and
increased yield (Kahlown et al., 2005; Mejia et al., 2000). In the Amazonian arc of deforestation, irrigation of
crops is relatively uncommon  (Lathuillière et al., 2012) and increases in agricultural productivity have been
achieved primarily by increasing the area of crops (Oliveira et al., 2013). If agricultural vegetation can access soil
moisture in these shallow WTD areas it could potentially increase the growing season length and productivity
without the need for investment in irrigation systems. In turn, less land would be required to achieve the same
agricultural output. During the wet season, soybean can reach rates of evapotranspiration similar to that of forest
(Costa and Foley, 2000). Some studies have suggested that the difference in annual ET between forest and
agricultural crops is primarily due to access to water during the dry season (Costa et al., 2007).

In this study, we use a number of freely available remote sensing products in combination with modelled water
table depth to investigate if naturally occurring shallow water table depth could increase evapotranspiration
compared to deep water table depth. We expect the greatest influence to be seen in crop areas as they have the
shallowest rooting depth and are most dependent on precipitation. As reported in other studies the influence of
WTD should not be visible for deep rooted vegetation (Nepstad et al., 1994) like forest and some savanna species.
As savanna has mixed vegetation and rooting depths, we expected to find some differences in ET as a result of
deep and shallow WTD. We expect that the differences as a result of WTD will be greater in the transition periods
between wet and dry seasons as rainfall as a water source is limited. In areas of shallow WTD, the saturated zone
is closer to the root zone of vegetation. In these locations we, therefore, expect crop vegetation to be buffered
against the reduction in rainfall during the dry season transition and experience drought conditions later, thus
delaying the decline of transpiration due to the dry season. Similarly, during the wet season transition, we expect
that areas of shallow WTD will have higher productivity as crop vegetation may access the shallow WTD to
supplement their demand when rainfall is low, therefore growing sooner than areas with deep WTD, effectively
shortening the dry season. Finally, we discuss whether differences found in ET between deep and shallow WTD
are important for moisture recycling, vegetation productivity and what are the implications for future LULCC.

## 2 Methods

### 2.1 Study Area

The study area is located in the southern Amazon, mostly in the northern region of Mato Grosso and incorporating the border area with Pará (Figure 1). Mato Grosso is classified into three major biomes with rainforest in the North, cerrado (a vegetation type that resembles savanna) in the central region and wetlands in the southwest (Kastens et al., 2017; Lathuillière et al., 2012). The climate has two seasons, the wet season in the austral winter and the dry season in austral summer, the dry season lasts around 5 months with an annual average rainfall of 2000 mm and monthly mean temperatures between 22 - 26 $^\circ$C (Arvor et al., 2014). This precipitation level is within the natural range supporting both savanna (700 to 2000 mm/year), and forest (1000-2500 mm/year). Mean elevation over the study area is 345 m $\pm$ 100 m with a maximum of 700 m and a minimum of 100 m. Runoff in the Amazon basin is usually low with groundwater convergence accounting for as high as 90% of streamflow (Miguez-Macho and Fan, 2012). Mean WTD of the study area is 12 m with approximately 20 % shallow (< 2 m). The maximum WTD is 60 - 70 m. This region is well known as a dynamic agricultural frontier – the arc of deforestation – with high rates of LULCC, where forest and savanna are converted for extensive agriculture, mostly cattle ranching and soy production (Kastens et al., 2017). Mato Grosso is the leading producer of agricultural crops such as soybean in Brazil (Gusso et al., 2014). We chose a 750 km x 750 km study area which is centrally located in the arc of deforestation and has large areas of primary forest (73 %), savanna (19 %) and crops (3 %).

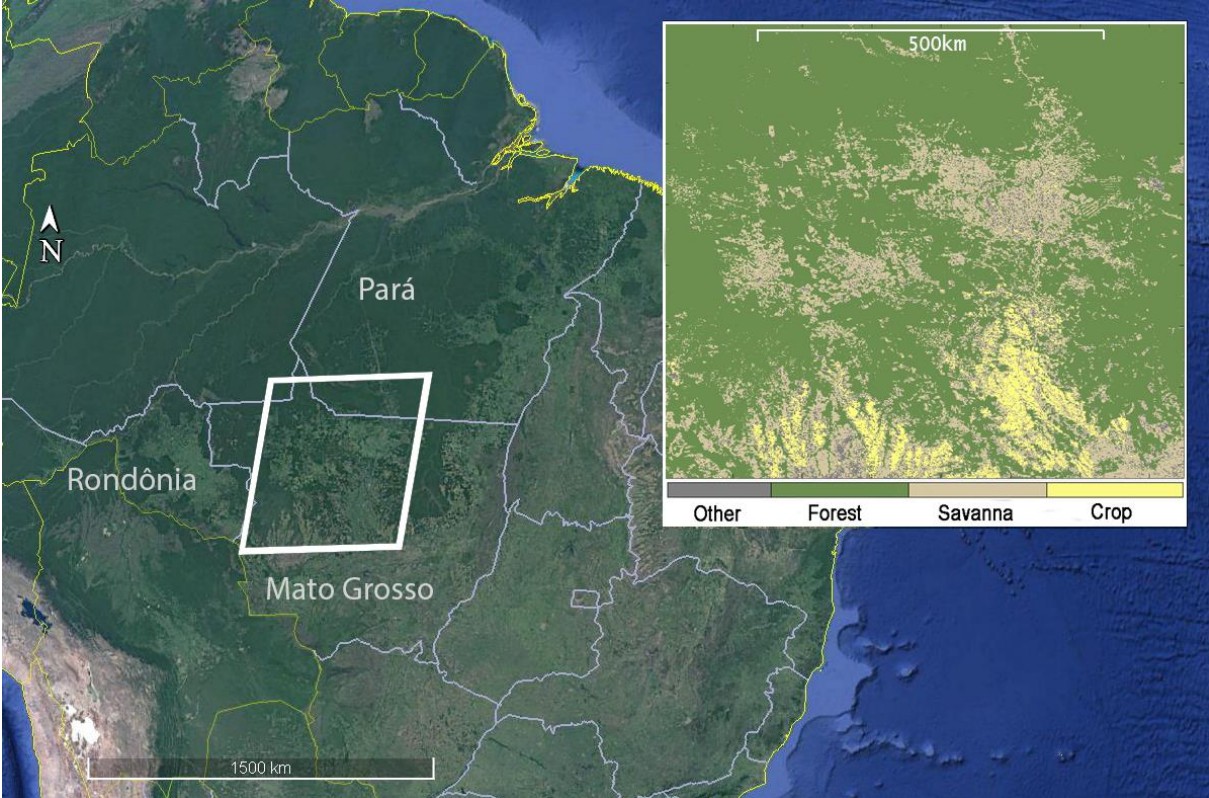

**Figure 1: Study area on the arc of deforestation the Amazon, in Northern Mato Grosso. Inlayed image shows MODIS land cover classification map (2001) for the three land cover classes analysed. Forest – Green, Savanna – Beige, Crop – Yellow and Other - Grey. Due to the sinusoidal projection of MODIS satellite data, the study area looks distorted. Satellite image data: Google Earth, Landsat / Copernicus**

**2.2 Datasets**

**2.2.1 Remote sensing data**

Remote sensing offers excellent tools for monitoring changes in vegetation over large regions as it provides full geographic coverage, high temporal frequency at spatial scales relevant to most Earth system processes (Chambers et al., 2007). Here we use three separate products from the Moderate Resolution Imaging Spectrometer (MODIS), namely MODIS Evapotranspiration (MOD16A2), MODIS Land Surface Temperature (MOD11A2), and MODIS Enhanced Vegetation Index (MOD13A2), to assess the influence of WTD on evapotranspiration. MODIS remote sensing products were used as they offer a moderate spatial resolution and a high temporal resolution which is ideal for examination of seasonal processes. We chose to perform the analysis for the currently available MODIS land cover archive using data from 2001 to 2012. In addition, this period represents a time with high variability of precipitation extremes in which the Amazon experienced droughts, floods and could depict the variability the system experiences (Nobre et al., 2016). Data was downloaded from the NASA data sharing portal (earthdata.nasa.gov). Data was rescaled to 1 km resolution, no additional post-processing was conducted.

MODIS Evapotranspiration (hereafter ET) data (Mu et al., 2011) provides 8 day accumulated evapotranspiration at 500 m resolution (rescaled to 1 km). The ET dataset is one of the best available datasets due to its relatively high spatial and temporal resolution as such it has been widely used to investigate the effect of land use change on evapotranspiration in the Amazon (Loarie et al., 2011; Neill et al., 2013; Vergopolan and Fisher, 2016). The baseline algorithm to the MODIS ET product is based on the Penman-Monteith equation, and combines parameters such as land cover, leaf area index (LAI), Albedo and Fraction of Photosynthetically Active Radiation (FPAR) directly observed with or modelled from MODIS data, with reanalysis data on Radiation, Air Temperature and Humidity from the Global Modelling and Assimilation Office (Mu et al., 2011). The MODIS ET products were previously tested over the Amazon by comparing its outputs with eddy covariance tower data, showing that the product is more accurate over longer temporal scales (monthly timesteps) and larger areas (e.g. drainage basin) (Ruhoff et al., 2013; Velpuri et al., 2013). While MODIS ET product is known to be underperforming at fine temporal resolutions and newer novel methods show promising results at nine flux sites across the Amazon (Xu et al., 2019), we believe that the application of the new method for our question on the influence of WTD and our time series analysis was beyond the scope of this study. This is also the reason why we chose to also analyse the effects of WTD on satellite retrieved EVI and LST. As with these additional products differences might be detectable, and potentially show a signal to the effects of WTD on the water cycle.

MODIS Land Surface Temperature (hereafter LST) provides an 8 day mean day time land surface temperature in degrees Kelvin at 1 km resolution. LST data are produced by detection of thermal infrared radiation between 3 – 15 µm spread across 15 bands of the thermal sensor on board the MODIS satellite system and temperatures are modelled based on land cover classification with a clear sky accuracy of 1 degree K (Wan, 2014). MODIS LST data was converted to degrees Celsius. Evapotranspiration in the Amazon has been shown to result in a net cooling effect (Bonan, 2008) therefore, areas with lower LST will be observed in areas of higher ET (Eltahir and Bras, 1994).

MODIS Enhanced Vegetation Index (hereafter EVI) provides an observation on vegetation greenness at a frequency of 16 days and 500 m resolution (rescaled to 1 km). EVI is a vegetation index that measures greenness as a proxy for productivity (Huete et al., 2002). It was developed to improve upon the normalized difference

vegetation index (NDVI), as EVI is less sensitive to saturation in highly dense canopies as those in the Amazon, and EVI also corrects for canopy background effects and atmospheric aerosol effects (Huete et al., 2002). This MODIS product offers an observation of vegetation productivity as it measures "greenness" and is correlated to photosynthesis/evapotranspiration (Mu et al., 2011). Thus vegetation with adequate access to water near their root zone will have a comparatively higher EVI than vegetation which is water stressed. This higher EVI, in turn, would correspond to areas of higher ET.

In addition, we also used the MODIS land cover product for selection of our analysis sites (see below). MODIS land cover (hereafter land cover) provides a classification of global land cover at 1 km resolution, and it is annually updated. For this study, we only used pixels that were classified as the same land cover type during the entire study period 2001 - 2012. The study area chosen provides a sufficient number of representative pixels for random selection of each land cover type. The use of stable land cover classes was necessary to determine and describe the patterns of ET, LST, and EVI over time and assess the effects of WTD on such trends without the confounding effect of land cover change. Further, we used MODIS land cover as it is the same land cover classification map as used for the MODIS ET product (Friedl et al., 2010) to avoid effects of land cover classification errors from different maps.

Over the Amazon cloud cover and shadows are an issue, especially in the wet season. Pixels with high cloud cover were excluded from the analysis. The high seasonal difference in cloud cover is clear, at each time step we used a spatial mean of only available pixels, due to our large sample size we still have enough pixels for the analysis (see figure SI.10.1). We compared the cloud cover per land cover class, and found no bias or significant differences between deep and shallow areas.

Topography might influence the MODIS data in an number of ways. Elevation can influence meteorological forcing (i.e. temperature and vapor pressure) which is used to calculate ET. Topography can also influence water availability on a pixel due to slope and catchment size of the surrounding area, impacting water available to vegetation therefore influencing ET and EVI. Serious errors due to topography are filtered by MODIS quality control dataset and these pixels were excluded from our analysis. We used SRTM (Shuttle Radar Topography Mission) data to examine elevation and calculate the topographic wetness index (an integrated measure of water accumulation) of our studied pixels. No significant differences were found between elevation of deep and shallow WTD areas of forest or savanna. Crop elevation was found to be significantly different between deep and shallow areas for half of our randomisations. However, the difference in mean elevation was only 10 m leading us to believe that this will not have a strong impact on the meteorological forcing data or ET. We found no significant differences in the topographic wetness index between deep and shallow land covers (see figure SI.9.4).

Finally, Tropical Rainfall Measuring Mission (hereafter TRMM) 3B42 provides daily precipitation at a resolution of 0.25 $^0$ (downloaded from earthdata.nasa.gov). We calculated daily mean rainfall of our study area using the TRMM data which was then used to calculate the seasonality of rainfall, i.e. start of the dry season and the wet season across the study area and not per pixel (see below for further details).

**2.2.2 Water table depth**

Water table depth (WTD) values were extracted from the Fan et al. (2010) equilibrium WTD model of South America at 30 arc seconds (~ 1 km). The model was created as a long term mean water table depth using a combination of literature reported depths and national databases of groundwater table depth most of which are

from drinking water wells from areas of high population. This data is interpolated using a groundwater model
forced by present day climate, terrain, and sea level.. We used the output of the model to obtain WTD data, which
was projected to the same sinusoidal projection of the MODIS data. The equilibrium WTD model is intended for
use in dynamic simulations, and although our study is not the intended use of the WTD model, it is the best
currently available.  As the WTD model output is in "equilibrium" it gives a better indication of the annual average
WTD compared to interpolated WTD measurements which may be biased depending on when they were recorded.
The authors compared their WTD calculations with values reported in the literature and found good agreement for
shallower WTD; however, the model overestimated deep WTD. We selected two broad WTD classes in order to
further reduce some of the uncertainty around this key parameter: Shallow <2 m and Deep >8 m (and we will refer
to these as such from hereafter). Figure 2. shows a theoretical graphical representation of the difference between
forest (deep rooting depth), savanna (mix rooting depth), and crop (shallow rooting depth) land cover classes.
These depths were selected as they represent rooting depth values for crop and forest vegetation from literature
(Fan et al., 2016; Moreira et al., 2000; Nepstad et al., 1994; Setiyono et al., 2008).
**2.3 Sampling design**
**2.3.1 Spatial sampling**
We chose to avoid pixels which experienced LULCC during the study period as we wanted to use the full time
series for each pixel. We used MODIS land cover to identify pixels of each land cover class which remained
unchanged between years and used these for analysis. We combined three land cover classes with the two water
table depths and analysed the following classes: Forest Deep, Forest Shallow, Savanna Deep, Savanna Shallow,
Crop Deep, and Crop Shallow.
For each class, we randomly selected 1000 pixels and performed this random selection 20 times to account for the
effect of the randomization process in the results. This random selection method increased computational
efficiency by limiting the number of total pixels examined and producing comparable group sizes for statistical
analysis. During the wet season the number of usable pixels was as low as 200 – 300 pixels per class for some
time steps while in the dry season the number of usable pixels was above 900 (see supplemental information fig
SI.8.1).

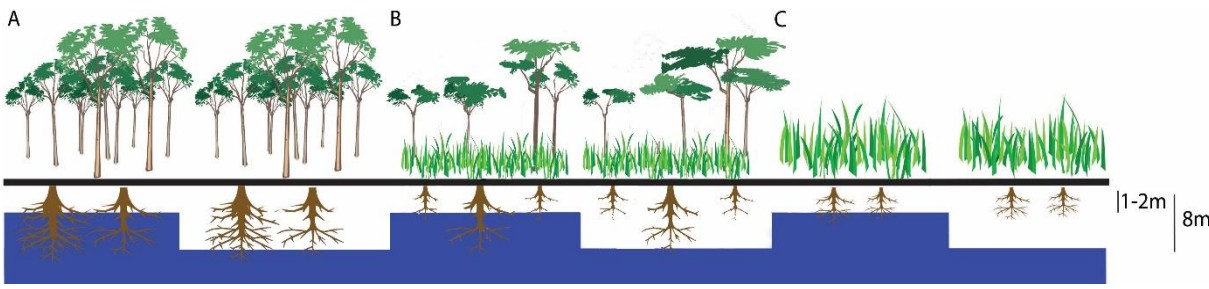


**Figure 2: Diagram showing that forest (A) root depth can reach until the saturated zone in both shallow (< 2 m) and**
**deep (> 8 m) WTD, savanna (B)  has a mixed rooting depth with only tree roots reaching deep WTD and  crop (C)**
**vegetation has a low maximum rooting depth (crops having a maximum rooting depth of 2 m and savanna having a**
**maximum rooting depth > 10 m (Canadell et al., 1996). Shallow roots can reach the saturated zone in shallow WTD (<**
**2 m); however, they cannot reach the saturated zone in deep WTD (> 8 m).**

### 2.3.2 Data analysis

The Amazon arc of deforestation is located in a region that has two major seasons defined by the difference in rainfall, the wet season from October to March (approximately 1500 mm) and the dry season from April to September (approximately 400 mm). The difference in rainfall can have significant impacts as the area can be prone to both seasonal flooding and droughts. In recent years the Amazon arc of deforestation has undergone an increased frequency of extreme weather events with drought in 2005, 2010 and flooding in 2009, 2012 (Nobre et al., 2016). These extreme climatic conditions can have a large influence on ET, and vegetation distribution as waterlogging of soils can lead to anoxia in the root zone. Due to the selection of only consistently classified pixels the influence of waterlogging can be avoided as over time these areas will fall under different classifications. Investigation into the drivers of these extreme variations and how each land cover class is influenced is however beyond the scope of this study.

Analysis of the data was conducted using three primary time periods. We compared mean daily values of ET, EVI and LST between deep and shallow WTD as this gives an indication of the influence of WTD on our land cover classes without considering the seasonal variation. We then compared ET, EVI and LST of our land cover classes during the dry season transition (DST) and wet season transition (WST) periods.

For each year we calculated the DST and WST using mean daily precipitation of our study area from TRMM with the anomalous accumulation method of Liebmann et al., (2007). This method uses the following equation:

$$A(n) = \sum_{n=1}^{day} [R(n) - \bar{R}]$$

Where $R(n)$ is daily precipitation and $\bar{R}$ is the average daily precipitation. Calculation of the anomalous accumulation begins at the driest month of the year, when the difference between daily precipitation and annual average is summed to a running total of the anomalous accumulation ($A$). The wet season onset is defined as the beginning of the longest period where the anomalous accumulation remains positive while the dry season onset is defined as the day after this anomalous accumulation reaches its maximum (Figure 3). These onset points of the dry and wet seasons were applied to find the closest time stamp from each MODIS product in the time series. We then considered the DST to last on average 8 repeats in the MODIS record (5 for EVI due to the lower frequency of the product) and the WST 7 repeats (4 for EVI). We used an average value for each remote sensing product over these transition periods to assess the difference between shallow and deep WTD on evapotranspiration.

302

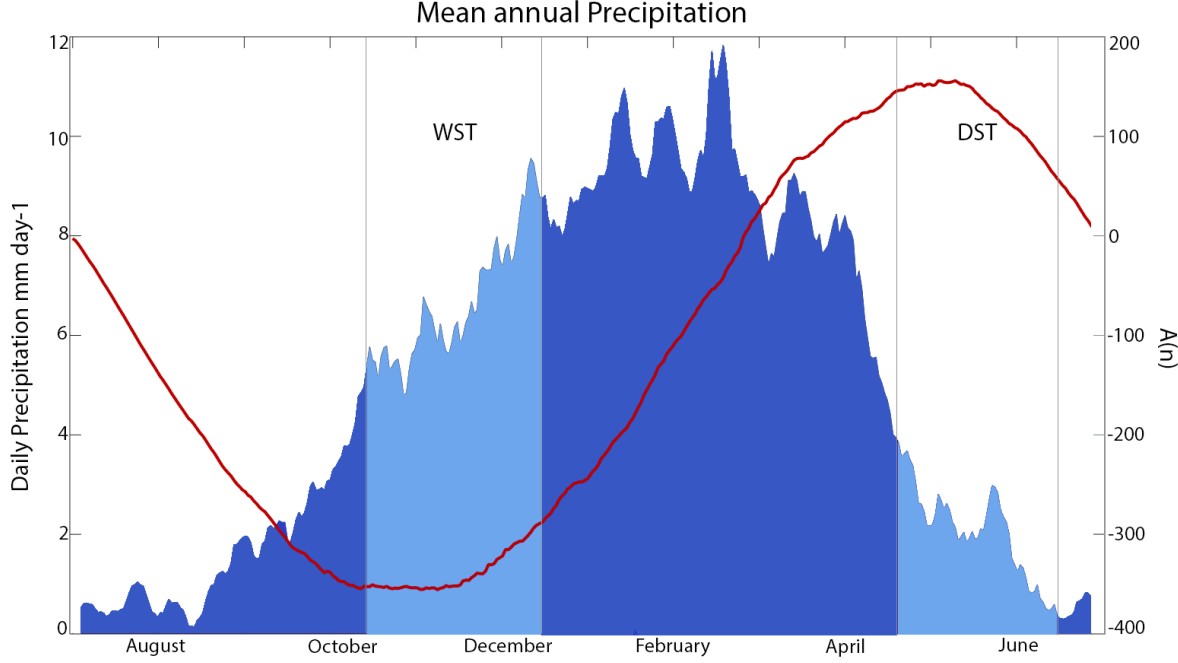

303

**Figure 3 Mean annual precipitation of the study area calculated from TRMM with 7 day average for graphical smoothing. wet season transition (WST) and dry season transition (DST) periods are represented in lighter blue. The vertical lines represent the average start and end dates, however exact dates were calculated per year between July 2001 and July 2012. Red line represents anomalous accumulation method A(n) from Liebmann et al. (2007).**

The WST and DST periods were selected as LULCC in the arc of deforestation is correlated with a lengthening of the dry season in particular delays in the WST (Butt et al., 2011; Dubreuil et al., 2012; Fu et al., 2013). Recently, evapotranspiration has been shown to draw moist air over the Amazon triggering the wet season before migration of the ICTZ (Wright et al., 2017). In this study, we focus our analysis on differences in the DST and the WST. During the DST, there is already significant drydown (anomalous accumulation is at a maximum, and precipitation already went down before, see figure 3) which should be apparent in vegetation without access to deeper water sources. Further into the dry season, other factors may cause a decline in transpiration as well, like heat stress. During the WST, we focus on the recovery of the vegetation, which should be faster when they have access to deeper water sources, like deep roots or a shallow WTD. Thus shallow rooted vegetation in shallow WTD areas may have higher access to water as their root zone is closer to the water table this will likely produce higher ET, EVI and lower LST during the DST than shallow rooted vegetation in deep WTD areas. This is because the WTD is much deeper and further from the vegetation rooting zone, which leads to a lack of access to water and the vegetation will likely be stressed. Similarly, during the WST, shallow rooted vegetation in shallow WTD may exhibit higher ET, EVI and lower LST than that in deep WTD because vegetation cannot yet be sustained by precipitation alone.. We do not expect these differences with deeply rooted vegetation

We tested whether ET, LST, and EVI followed a normal distribution using the Kolmogorov–Smirnov test. This test served two purposes, to assess whether parametric statistics could be used and also indicate whether the WTD influences the frequency distribution of ET, LST, and EVI. Since a large number of response variables were not normally distributed, we chose to use non-parametric methods. Therefore, Wilcoxon rank sum test was used to

test whether there was a significant difference in median ET, LST, and EVI due to the deep and shallow water
table.
We further examined the frequency distribution of deep and shallow WTD of each of the datasets using the
methodology of Wilcox (2012) where the lower and upper quantiles of the distribution are compared. Wilcox's
method utilises bootstrapping in order to compare the distribution of the 10th and 90th quantile using the Wilcoxon–
Mann–Whitney test. Due to our large sample size, 100 bootstrapped datasets were used.
Statistical analysis between each deep and shallow land cover pair was performed separately each year for all 20
randomisations e.g. differences in forest ET was tested for significance 12 years * 20 randomisations. For one
year, the difference in ET, EVI or LST was considered statistically significant when more than 66.7% of
randomisations were significant and an overall significance was determined if the majority (>50 %) of the years
were significant. Statistical analysis was performed using Matlab R2018a (The MathWorks Inc., Natick, USA)
statistical toolbox and Wilcox (2012) quantile distribution tool.
**3. Results**
The following results section is split into three subsections, one for each of the MODIS products used in the
analysis. Each of the subsections and accompanying figures follows the same structure. Each figure uses three
panels for the three time periods on the analysis A) annual daily mean, B) daily mean during DST, C) daily mean
during WST. Each panel has three pairs of box plots which represent the deep and shallow WTD data for forest,
savanna and crop.
**3.1 Effect of ground water depth on Evapotranspiration**
None of the three land cover classes had significant differences in the average daily evapotranspiration ($ET_{daily}$)
between deep and shallow WTD areas (Figure 4A). However, while we did not find consistent significant
differences, in both forest and crop $ET_{daily}$ we do see a trend towards higher $ET_{daily}$ in shallow WTD areas for both
(average ± standard deviation: Forest Deep = 3.953 ± 0.08 mm day$^{-1}$, Forest Shallow 3.967 ± 0.09 mm day$^{-1}$; Crop
Deep = 1.697 ± 0.07 mm day$^{-1}$, Crop Shallow= 1.713 ± 0.08 mm day$^{-1}$). Interestingly, we found significant
differences for Savanna at the extremes of the distributions, depicted by the arrows in Figure 4A. Both the 10th and
90th quantiles of $ET_{daily}$ were significantly higher in deep WTD areas than in shallow (difference of 10th = 0.017mm
day$^{-1}$, difference of 90th = 0.02 mm day$^{-1}$, see supplemental information table S.2.4 for all the quantile analyses).

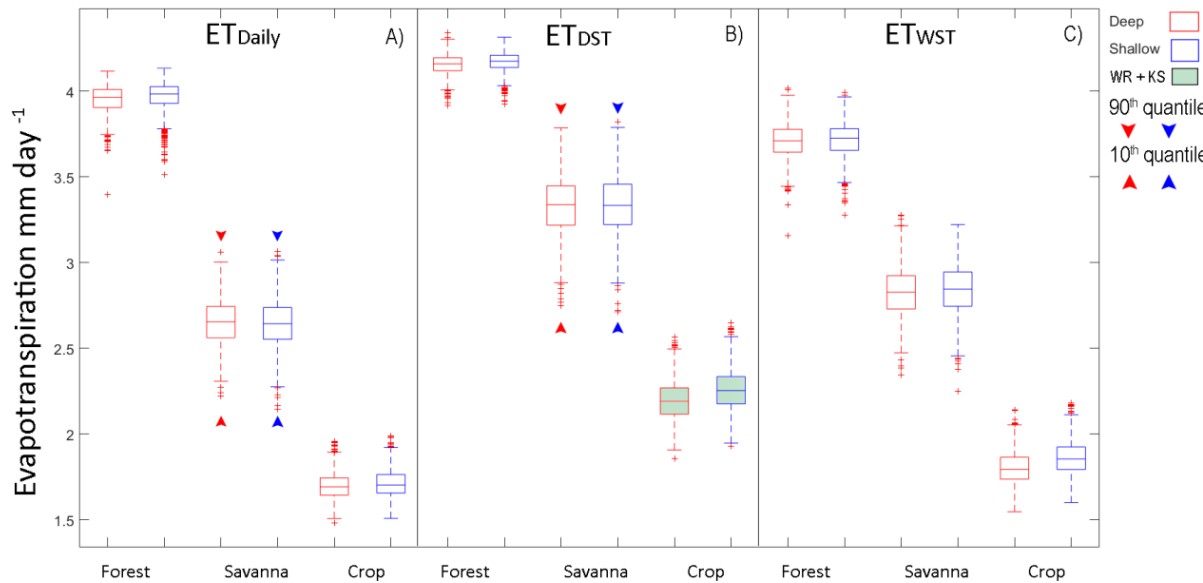


**Figure 4. (A) Average daily evapotranspiration (ET) annually ET_daily, (B) during the dry season transition period ET_DST,**
**(C) during the wet season transition ET_WST. Red boxes represent deep WTD Blue boxes represent shallow WTD.**
**Significant results are shown by the green filled boxes if significance was found with both Wilcoxon Rank (WR) and**
**Kolmogorov–Smirnov (KS). Significant differences in 10th and 90th quantile are depicted by the arrows.**


Clear differences in seasonality occur between the different land cover types (see supplemental information figures
SI.3.1, SI.3.2 and SI.3.3). During the wet season mean ET of all land cover types can be above 4 mm day$^{-1}$. Both
crop and savanna show clear suppression of ET during the dry season.

Crop ET during the DST (hereafter ET_DST) was significantly higher in shallow than deep WTD areas (average ±
standard deviation ET: Deep = 2.196 ± 0.11 mm day$^{-1}$, Shallow = 2.26 ± 0.12 mm day$^{-1}$, see the green filled boxes
in Fig 4B). Again we observed significant differences at the extremes of the distribution for savanna, on average
the 10th quantile of ET_DST was higher in shallow (average difference = 0.003 mm day$^{-1}$) and on average the 90th
quantile of ET_DST was higher in shallow (average difference = 0.005 mm day$^{-1}$).
ET during the WST (hereafter ET_WST), while on average ET_WST was higher in shallow WTD areas than in deep
WTD areas (average difference: Forest = 0.01 mm day$^{-1}$; Savanna = 0.01 mm day$^{-1}$; Crop = 0.06 mm day$^{-1}$) this
difference was not significant (Figure 4C).

**3.2 Effect of ground water depth on Land Surface Temperature**
We found that the distribution of the average land surface temperature (LST_daily) was significantly different only
for savanna and the 90th quantile of crop. Deep WTD areas of savanna showed a distribution skewed towards lower
temperatures (average ± standard deviation LST: Deep = 31.705 °C ± 0.38, Shallow = 31.848 °C ± 0.37), see yellow
filled boxes in Figure 5A. The 90th quantile of crop LST_daily deep WTD areas was on average 0.1 °C higher than in
shallow WTD areas. Although this is only part of the distribution, it indicates that the warmest crop areas are found
in deep WTD.


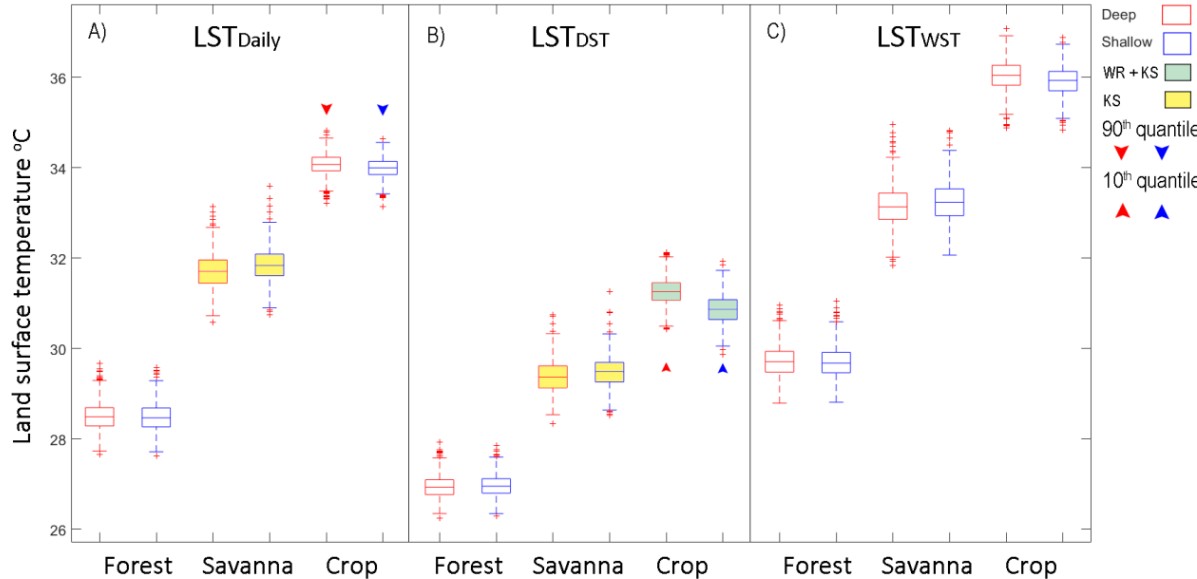


**Figure 5. (A) Average daily land surface temperature (LST) annually LST$_{daily}$, (B) during the dry season transition period LST$_{DST}$, (C) during the wet season transition LST$_{WST}$. Red boxes represent deep WTD Blue boxes represent shallow WTD. Yellow filled boxes represent a statistical difference in skewness, calculated by Kolmogorov–Smirnov, and green filled boxes represent statistical differences by both Wilcoxon-rank and Kolmogorov–Smirnov. Significant differences in 10$^{th}$ and 90$^{th}$ quantile are depicted by the arrows.**

389

LST shows clear seasonal differences between the different land covers. Crop LST has the highest range in LST with the warmest period coming towards the end of the dry season (August/September). (Supplemental information figure S.5.1, S.5.2 and S.5.3). During the DST, we found that crop in deep WTD areas had a significantly higher LST than in shallow WTD areas (average ± standard deviation LST: Deep = 31.256 ± 0.29 °C, Shallow = 30.864 ± 0.31 °C, green filled boxes in Figure 5B). In addition, the 10$^{th}$ quantile of the crop distributions was significantly higher by 0.42 °C in deep WTD areas than in shallow. During these periods we found again a significant difference in the distribution of savanna, where deep savanna distribution was skewed towards lower LST values. No significant differences were found during the WST (Figure 5C).

**3.3 Effect of ground water depth on Enhanced Vegetation Index**

We found significant differences in daily average EVI (EVI$_{daily}$) between deep and shallow WTD only in crop (average ± standard deviation EVI: Deep = 0.352 ± 0.01; Shallow = 0.357 ± 0.01), with shallow WTD areas EVI being higher than that of deep WTD areas (Figure 6A green filled boxes).

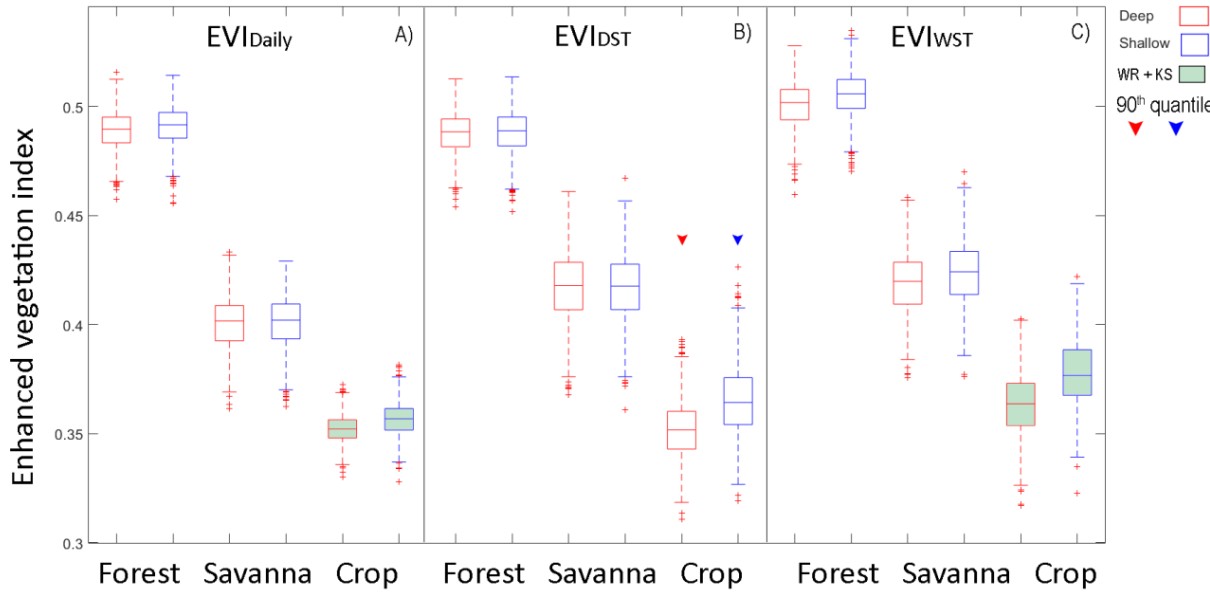

403

**Figure 6. (A) Average enhanced vegetation index (EVI) annually EVI$_{daily}$, (B) during the dry season transition period EVI$_{DST}$, (C) during the wet season transition EVI$_{WST}$. Red boxes represent deep WTD Blue boxes represent shallow WTD. Green filled boxes represent statistical differences by both Wilcoxon-rank and Kolmogorov–Smirnov tests. Significant differences in 90$^{th}$ quantile are depicted by the arrows.**

408

Seasonality in EVI is shown in Supplemental information figure S.7.1, S.7.2 and S.7.3. Crop EVI shows the highest variation among land cover types. When looking at the DST (May/June) of crop EVI it seems that the response is delayed in shallow WTD compared to deep WTD; for the WST (October/November) it seems that EVI in shallow areas increases faster than in deep WTD areas.

Mean EVI during the DST (EVI$_{DST}$) for crop showed a trend towards higher EVI in shallow WTD areas; however, this difference was only significant in 5 of the 11 years and therefore is not considered consistent enough to be statistically significant (average ± standard deviation EVI: Deep = 0.352 ± 0.01, Shallow = 0.3656 ± 0.01. Figure 6B, Table S.6.8). The 90$^{th}$ quantile EVI of crop was significantly higher in shallow WTD areas than deep. During the WST (EVI$_{WST}$), crop was the only different class where EVI was significantly higher in shallow WTD areas that in deep WTD areas (average ± standard deviation EVI: Deep = 0.364 ± 0.01, Shallow = 0.378 ± 0.02, green filled boxes in Figure 6C).

## 4 Discussion

In this study, we tested the hypothesis that areas of shallow water table depth (WTD) would have higher evapotranspiration when compared to areas of deep WTD. As crop vegetation has the shallowest roots (< 2 m) we expect to see the largest influence of WTD in crop vegetation. In areas of deep WTD the root zone is far from the saturated zone resulting in less uptake of deep soil water, while in areas of shallow WTD the root zone is close to the saturated zone therefore providing the crops access to ground water. However, we found no support for this as the annual daily mean ET was not different between crop in deep and shallow WTD areas. One potential explanation is that since crops experience high seasonality, this annual variability may override differences

between deep and shallow WTD areas in the daily average values of ET. For example, average crop ET reaches a maximum of 3.5 mm day$^{-1}$ in the wet season while the dry season ET reaches a minimum of 0.4 mm day$^{-1}$. Interestingly we found significant differences in annual mean LST and EVI for crop. For LST, we found that the upper 90[th] quantile was 0.11 °C higher in deep than in shallow WTD areas. While this difference is only found in the 90[th] quantile of the distribution it does indicate that LST in deep WTD areas can reach higher temperatures than shallow WTD areas. In addition, we found that crops in shallow WTD areas had a significantly higher EVI than in deep WTD. Crop EVI in shallow WTD areas is 1.2 % higher than in deep WTD. This provides support to our hypothesis that crop would have higher EVI in shallow WTD compared to deep WTD areas. The maximum rooting depth for most crops in the region is 2 m, in shallow WTD areas this means the root zone is close to the WTD and would have access to water while in deep WTD the roots are far from the saturated zone. This access to water in shallow WTD areas could also lead to higher ET and therefore evaporative cooling could explain the cooler temperatures in the 90[th] quantile.

The second part of our hypothesis was that the effect of WTD would be most evident during the transition periods between wet and dry seasons when rainfall is reduced and vegetation activity is limited by access to soil moisture. We found support for this hypothesis during the DST. In the DST, crop ET was significantly higher in shallow WTD areas and crop LST was significantly lower in shallow WTD areas, while in crop EVI we saw a trend towards higher EVI in shallow WTD areas (significant differences were only found in 5 of the 11 years). While the difference in crop ET is not large (0.063 mm day$^{-1}$, 2.9 % higher in shallow); during the DST, the results are important as they indicate that crops in the shallow WTD areas have a delayed response to lower rainfall and have a relatively longer growing season. Further evidence of this delayed response can be seen in the EVI seasonality graphs (see figure SI.7.3) where the response of shallow crop to the DST seems delayed compared to deep areas. Crop LST further supports our hypothesis as LST in deep WTD areas was 0.39 °C higher than in shallow WTD areas, while no significant effects were found in EVI. Therefore cooler temperatures in shallow WTD areas are expected to be the result of higher evaporative cooling from ET. These relatively low differences in ET as measured with MODIS data might also be due to the ET product itself. The ET model used for MODIS is not optimised for comparison over relatively small spatial extents and short temporal scales (Ruhoff et al., 2013). In addition, the ET model does not take into account soil water storage and ET is based largely on atmospheric forcing and global land cover parameterisation. Therefore the differences we found for the DST may be underestimated in the MODIS ET values.

Ponte De Souza et al. (2011) highlighted that one of the strongest impacts of LULCC from forest to crop was due the simultaneous 85% increase in sensible heat flux and 78% reduction in latent heat recorded during the dry season. Studies examining the change in LST due to LULCC found that LST increased by 6 °C from forest to crop (Silvério et al., 2015) and 1.5 °C from savanna to crop (Loarie et al., 2011). Further global models estimated an increase of 5 °C during the summer season for the Amazon, due to a shift from forest to grass (Brovkin et al., 2009; Dekker et al., 2010). This increase in temperature could be influenced by WTD and land cover change; in shallow WTD areas this may result in a less severe temperature change while in deep WTD it could lead to a greater change in temperature; however, WTD was not used as input for these modelling studies. Our results show a maximum temperature of 30 °C in forest compared to a maximum temperature of 38 °C in crops.


We also expected that the influence of WTD would be important during the WST, as in this period rainfall is
increasing. In areas of shallow WTD, vegetation with a root zone close to the water table may still access water to
supplement if rainfall is not sufficient. Therefore, vegetation growth may be accelerated in comparison to areas of
deep WTD which rely more directly on precipitation. Crop EVI was significantly higher in shallow than deep
WTD areas by about 3.8 %, and this was the only data for which we found a significant difference. Looking at the
seasonality of EVI (figure SI.7.3) during the WST EVI is increasing faster in shallow WTD areas than in deep
WTD. EVI measures vegetation greenness and could be an indication of more rapid growth in shallow WTD areas.
As EVI data is directly observed and not modelled the differences are solely reliant on differences in reflected
radiation. It may be that smaller differences between deep and shallow WTD areas are more easily detectable using
this data. Along the arc of deforestation observations of a lengthening dry season since the 1970s, are linked to a
delay in the WST (Butt et al., 2011; Fu et al., 2013). This delay correlates with LULCC and the large reduction
this has on ET (Debortoli et al., 2017). Although the difference in WTD seen in crops does not have a strong
influence on ET when compared to the difference in ET between the land cover classes, evidence of earlier or
faster growth due to the shallow WTD could be beneficial on a local scale.

These results are even more relevant when comparing the effects of WTD in crop and forest. As forest has been
shown to maintain ET throughout the seasons (Kunert et al., 2017) as its deep roots access deeper groundwater
(Gash and Nobre, 1997; Nepstad et al., 1994), we hypothesised that no change should be observed in ET, LST,
and EVI. Indeed, we found no significant differences across the three MODIS products, both annually or during
the DST and WST. While this does not directly support our hypothesis about the role of WTD for shallow rooted
vegetation, this does help validate that our approach reflects our knowledge of the system for vegetation with deep
roots.

Savanna is a complex land cover type because its natural structure makes it is challenging to classify with remote
sensing data (Gibbes et al., 2010). MODIS classification accuracy of savanna is about 40 %, about half of that of
forest and crop (90 % and 80 % respectively) (Friedl et al., 2010). Savanna includes both trees and grasses, which
through the deep roots of trees may access moisture directly and facilitate moisture uptake via hydraulic
redistribution (Oliveira et al., 2005) and large areas of shallow root grasses without trees would be negatively
affected by water stress. A number of the findings for savanna were not in line with our proposed hypothesis. The
distribution between shallow and deep LST was significantly different, with deep WTD areas having a skewed
distribution towards lower temperatures. In our hypothesis, we expected to find lower temperature where shallow
WTD occurs or no differences in temperature. A similar trend was found in ET where the $10^{th}$ and $90^{th}$ quantiles
of the distribution were significantly higher in deep WTD areas. The difference in ET was very small, less than 1
% difference between deep and shallow WTD areas. Water logging of soils has been shown to be an important
factor in determining vegetation distribution (Ridolfi et al., 2006; Rossatto et al., 2012). Although we believe that
larger flooding event leading to changes in vegetation composition are removed from our study due to the selection
of pixels that during the time series were always classified as one land cover type, shorter periods of water logging
may occur in shallow WTD areas. However, much higher spatial and temporal resolution imagery would be needed
to identify this possibility.

The differences found for crop support our hypothesis that shallow WTD areas may facilitate water uptake
compared with areas of deep WTD during the transition between wet and dry seasons. Previous crop production
studies have shown that artificially maintaining a shallow WTD through sub irrigation systems can increase the
productivity of crops such as soy (Kahlown et al., 2005; Mejia et al., 2000) but this has not been previously shown
in the naturally occurring shallow WTD areas of the arc of deforestation in the Amazon. In deep WTD areas, crop
vegetation undergoes more severe water stress compared with shallow WTD further reducing evapotranspiration
and its potential impact on the moisture recycling system. At the regional scale, the difference between deep and
shallow WTD is not that important. The most significant differences in ET are driven by deforestation and strong
annual variations in rainfall. Although not analysed specifically in this study, the remote sensing data clearly shows
these distinctions between different land cover classes and high seasonal and inter-annual variability. On a local
scale, the difference between deep and shallow WTD on crop may be of great importance. During the DST crop
areas in shallow WTD maintained higher ET. This difference may be important for overall productivity as the dry
season influence is delayed and as a result, is increasing the growing season length. This could facilitate natural
double cropping systems without the need for investment in irrigation which is still an uncommon practice in the
Amazon arc of deforestation (Lathuillière et al., 2012). Agricultural intensification is a pathway to increasing the
sustainability of agriculture in the arc of deforestation if it prevents or reduces deforestation or facilitates
reforestation (Oliveira et al., 2013). If agricultural productivity can be increased by focusing on already cleared
shallow WTD areas, areas of deep WTD could be reforested or returned to secondary forest. Reforestation of
previously degraded or logged forest has been shown to return to near natural levels of ET within a few years
(Davidson et al., 2012; Hölscher et al., 1997). The patterns seen in crop vegetation may be caused by factors not
considered in this paper. Spatially explicit details about specific crops or agricultural practices were not known for
the study. Planting of soybean is determined by the WST and can vary between September and October (Gusso et
al., 2014). It is possible that the differences seen in shallow WTD could be the result of earlier sowing and double
cropping systems. However, it may be that these agricultural management decisions are implemented more often
in shallow WTD because of the higher availability of soil water.

This study is a first approach into gaining a better understanding on the influence of shallow WTD on shallow
rooted vegetation and it heavily relies on models and remote sensing data which are most appropriate for analyses
at larger spatial and temporal scales.
The results presented here are limited by the inherent uncertainty of the data used, both in the WTD model and the
remote sensing data. Although we believe that the WTD model used here is the best currently available, due to
limited data availability it was created using data located mostly in the coastal regions of the continent with very
few observations from near our study site (Fan and Miguez-Macho, 2010). In this study, the authors note that there
is an overestimation of deep WTD areas when validated against literature reported values. We believe that by the
use of a conservative definition of deep WTD >8 m the model outputs are appropriate for our purposes. As
discussed above, the remote sensing data has obvious limitations but does provide some insights into how depth
of the water table at a local scale might affect water transfer and evaporative processes. Nonetheless, the second
main source of uncertainty is in the MODIS land cover classification. We chose to use this land cover classification
as the ET and LST products use this classification in their algorithm. Although the classes used are broad and do
not reflect the full complexity and heterogeneity of the arc of deforestation, they are robust enough for our
purposes. As the influence of WTD on ET is most relevant on smaller scales, further research in these areas could
focus on the smaller spatial scales and validate sites with accurate classification and WTD measurements.

**5 Conclusion**
This study aimed to investigate if naturally occurring shallow water table depth supported higher ET compared to
deep WTD. In particular if shallow rooted crop vegetation would have higher ET due to increased access to soil
water in shallow WTD areas as the distance from the root zone to the saturated zone is shorter. Comparison of EVI
showed evidence to support this hypothesis as daily mean EVI was significantly higher in shallow WTD crop
areas. However, the difference between deep and shallow WTD is overshadowed by the clear differences between
land cover classes. Although not the focus of this study, differences in ET, LST and EVI were largest between
land cover classes. In terms of larger scale processes like moisture recycling, LULCC is far more impactful than
WTD differences. The main driver of LULCC is agricultural expansion. So although our results are not directly
relevant at regional or continental scales on a local scale shallow WTD areas are more productive than deep WTD.
The influence of WTD on crop vegetation was concentrated during the transition periods between wet and dry
seasons. We found higher ET and lower LST during the DST and higher EVI during the WST for crop in shallow
WTD areas. This higher vegetation productivity of crops due to the shallow WTD help effectively increases the
growing season length. The higher productivity in shallow WTD areas may facilitate natural double cropping
increasing the agricultural efficiency of the areas. These local scale effects can become significant when scaled to
the level of the Amazon. Deforestation rates grew as high as 28,000 $km^2$ $year^{-1}$ in 2004 (Davidson et al., 2012).
Any LULCC which occurs in areas of deep WTD are leading to inefficiencies in agricultural production and higher
impacts to the moisture recycling system.
The results presented here help to demonstrate that the LULCC impacts can vary spatially due to differences in
WTD. Future studies investigating the impact of LULCC should incorporate WTD to help disentangle the full
impact on the moisture recycling system.

**Data Availability** https://search.earthdata.nasa.gov/ was used to access and download all MODIS, TRMM and
SRTM data used. Fan & Miguez-Macho's equilibrium water table depth for South America was downloaded from
https://glowasis.deltares.nl/thredds/catalog/opendap/opendap/Equilibrium_Water_Table/catalog.html

**Author Contribution** All authors contributed to the study in design and implementation. Original draft was
written by JOC, all authors contributed with revisions and editing.

**Competing Interest** The authors declare that they have no conflict of interest.

**Acknowledgements** This study was supported by funding from the graduate program 'Nature Conservation,
Management and Restoration' of The Netherlands Organisation for Scientific Research (NWO). We would like to
thank the reviewers for their helpful comments throughout the discussion process.

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
