# Peer review of "The influence of water table depth on evapotranspiration in the Amazon arc of deforestation"

_Hydrology and Earth System Sciences, 2019_

## Referee Comment (RC1) · Anonymous Referee #1 · 11 Mar 2019

O'Conner et al present an interesting and novel study on the effect of ground water depth (GWT) on evapotranspiration (ET), land surface temperature (LST) and the enhanced vegetation index (EVI). More precisely they study if the effect of GWT on ET, LST and EVI differs between vegetation types and season (with respect to rainfall). The authors find a strong difference in ET, LST and EVI between crop- and forest areas. Furthermore for crops they find a higher ET and lower LST in areas with shallower water table depth during the dry season transition.

The topic of the study is interesting and, as far as I know, they are the first to include the effect of GWT on ET at this scale. The set-up of the study (three land cover types, combined with two WTD classes) is easy to understand and effective. Figure 2 provides a good overview this set-up.

I have a few questions for the authors regarding the methodology (the used data sets and different choices made). Afterwards I wrote some general remarks on the content of the paper, followed by some suggestions with respect to the structure of the paper.

**Methodology**

- Three land cover types are studied: forest, savanna and cropland. The motivation for the inclusion of forest and cropland is clear: both are very different in structure and effect on the moisture recycling system. Also with respect to deforestation, these two land cover classes are a logical choice to study. The motivation for the inclusion of savanna is however not clear to me. What are the characteristics of the Cerrado savanna with respect to the water balance and what can be expected for this ecosystem? To make this more clear, I suggest to add root characteristics of savanna to figure 2.
- Three different time frames are studied: mean annual values, the dry season transition (DST) period and wet season transition (WST) period. The DST and WST are discussed in paragraph 2.3.2. Nevertheless it remains unclear to me why the WST and DST periods are selected, instead of the more extreme dry (and wet) season. Is it related to the planting and harvesting season of the crops?
- The authors selected the MOD16A2 data product to derive ET and briefly present the product as well as why this product is selected (one of the best available datasets, high spatial and temporal resolution, it is widely used). Also the authors describe that the remote sensing data has "obvious limitations" (L475). I have some concerns regarding this dataset and would like the authors to elaborate a bit on the characteristics and main limitations of using this MODIS data product in their study. Several studies validated the product (E.g. Velpuri et al., 2013) or wrote that especially for tropical sites across the amazon basin, the MODIS ET remains challenging (e.g. the recently published paper Xu et al., 2019).
- At the studied scale, the modelled water table depth classes are mainly based on the topography of the landscape. Are the MODIS products unbiased for this topography? E.g. is the LST corrected for topography and are the meteorological data required for MODIS ET calculation independent of topography?
- A few smaller points that unclear to me are:
  - Are the start and end of the DST and WST calculated for each TRMM pixel?
  - How many (cloudless) remote sensing data points are available? And is this enough to present the results (LST or ET values) with three decimal digits (e.g. L317, 329)?

**General remarks and questions**

- The results show that for cropland, EVI is higher for areas with a shallow WTD (paragraph 3.3 / L391). From the supplementary figures, it seems that deep WTD areas lag behind shallow

WTD croplands. Is this due to water conditions only, or could this be an effect of a different cropping regime? Do farmers adapt the species and timing of agricultural practices to the local conditions, e.g. length of the dry season?

- L184 "This MODIS product … is correlated to photosynthesis/evapotranspiration" (Sims et al., 2006). Please adjust this sentence, or add a reference (Sims et al., 2006 did not study evaporative fluxes).
- Caption figure 2: "while other vegetation has a lower maximum rooting depth". By other, do you mean crops and/or savanna? What is the rooting depth of savanna trees?
- A few lines are unclear to me:
  - L196 "Further, this choice avoids potential circularity in using land cover classification to detect an effect on a parameter that uses land cover classification to produce its modelled value"
  - L395 "if this extra warming above the canopy is caused by a change in ET, then better estimates of ET should be possible, however, this is not trivial"
  - L400 "therefore, the modelled data was expected to underperform, making the differences we found for the dry season even more important"

**Structure and writing**

I recommend to check the manuscript for spelling, punctuation and sentence structure. Below I give some suggestions that the authors could consider.

I recommend to more clearly differentiate between introduction, methods, results, discussion and conclusion. For example avoid hypothesis-like sentences in the methods section ("We expected that", L178), avoid discussion-like sentences in the results section (e.g. "as hypothesised", L316) and do not add new information to the conclusion. Personally I read the lines 490-502 like a discussion, instead of as the conclusion. Furthermore, I recommend to group the hypotheses in one paragraph and align these hypotheses with the discussion and / or conclusion, to guide the readers through the presented story. From the introduction I deducted four hypotheses and some of them are explicitly discussed, while one is not mentioned in the discussion. Also, some new (parts of) hypothesis are introduced in the discussion, which were not introduced earlier. E.g.

1. L102: "We hypothesise that areas of shallow WTD allow vegetation to access soil moisture, with both shallow and deep rooted vegetation potentially facilitating vegetation productivity and higher ET when compared to areas of deep WTD."
   - L369: "In this study, we tested the hypothesis that areas of shallow WTD would have higher ET when compared to areas of deep WTD, primarily in shallow rooted crop vegetation → Last part of this hypothesis is not mentioned in the introduction.
2. L116: "In areas of shallow WTD, the saturated zone is closer to the root zone of the vegetation. In these locations we, therefore, expect vegetation to be buffered against the reduction in rainfall during the dry season transition and experience drought conditions later, thus delaying the effect of the dry season". → This one is not (directly) referred to in the discussion
3. L428: "As forests has been shown to maintain ET throughout the seasons as its deep roots access deeper groundwater, we hypothesised that no change should be observed in ET, LST and EVI." → I didn't find this hypothesis in the introduction.

Some spelling related suggestions:

- L51 "changes (**reduction / decline**) in evapotranspiration reduce the available atmospheric moisture".
- L57 "forests can maintain a high rate of evapotranspiration during the dry season, **they are not** affected by low rainfall".
- L92 "agricultural vegetation … experiences high seasonality during the dry season unseen in forest vegetation". Seasonality in what?
- L130 "annual average temperatures ranging between 22 – 26 °C". Are 22 and 26 monthly mean temperatures?
- L167 "the MODIS ET products were previously tested … more accurate over longer temporal scales and larger areas". By this do you mean more accurate than shorter time/spatial scales?
- L180 "a 16 day repeat**ed** observation".
- L210 "and found good agreement **at/for** shallower WTD however,".
- L229 "these roots may **penetrate into the** soil until the saturated zone in shallow WTD; however, **do** not penetrate further in deep WTD".
- L240 "**three** primary time periods".
- L265 "we used an average value over these transition periods" (value of what?).
- L287 "a year was considered statistically significant". E.g. "for one year, the difference in .. was considered statistically significant".
- L302 "3.967 ± 0.0.09".
- E.g. L371/L379 "since crop experiences". I recommend to use for example "crop species" or "a crop" or "cropland".
- L377 "indicate that local conditions can be  warmer in deep WTD areas".
- L380 "the **roots** of crop vegetation only penetrates to a maximum of 2 m, in shallow..".
- L408 "this could mean that in deep WTD **areas** temperature could even be .."
- L409 "WTD **was** not".
- L444 "the difference in ET was very small, < 1% difference between deep and shallow **rooted areas**".

---

## Referee Comment (RC2) · Anonymous Referee #2 · 13 Mar 2019

General comments

In this paper, O'Connor et al tackle an interesting and very important question in the field of ecohydrology: how does groundwater affect plant functioning? As a community, it is important that we move from the broad, large-scale influences of climate towards focusing on the regional to local scales, where, as shown by several authors, groundwater might be one of the driving forces of ecosystems. This has important implications for our understanding of the response of natural and agricultural systems to climate change, and this study is a timely contribution to this field.

I believe, however, that there are some conceptual and methodological issues with this study that should be addressed before publication. Below I offer some comments on the content of the paper and also pose some questions that might help the authors in

further refining it.

Specific comments

1) The ultimate focus or "big question" of this study was somewhat unclear to me as I read the paper. In the introduction, a lot of importance is given to large scale problems such as the impact of land use changes on precipitation recycling and the subsequent negative effect on forest cover through a reduction in ET. However, in the conclusions section, the "key messages" are related to agricultural management and forest conservation. I believe the paper would greatly benefit from a clear, defined question that is posed in the beginning of the paper and that guides the discussion and conclusions. From the climate system point of view, the small differences in ET between shallow and deep WT observed in the study might not be significant, while from the perspective of sustainable agricultural management and general crop productivity these changes might suggest a more water efficient practice. Perhaps the authors could group their questions with their hypotheses, which currently are somewhat scattered throughout the introduction and methodology sections.

2) Although groundwater is the main environmental factor addressed in this study, very little is discussed about it throughout the paper. What is topography like in the region of study? How does the water table field look like in this area? How deep and how shallow can the water table be? What is the meaning of an "equilibrium water table depth"? What are the benefits and the drawbacks of using an equilibrium water table instead of a dynamic product? Is this an area where the water table responds directly to precipitation or is lateral convergence an important process? These are some key questions that directly impact the hypotheses and conclusion of this study, and therefore should be well addressed in the manuscript.

3) I don't understand the reasoning behind choosing the wet and dry season transitions as periods of stress for vegetation. The use of a climatic index neglects the important time lag displayed by groundwater (and soil moisture in general) that has been shown

to support considerable levels of ET well into the dry season for several places in the Amazon basin (Miguez-Macho and Fan, 2012). In fact, seasonal soil moisture storage maps from Miguez-Macho and Fan (2012) show that, in the top 2 m, October is a more critical month in this general area than the dry season transition (June/July) proposed here by the authors. Is there a specific reason for choosing these periods?

4) The authors should include early in the introduction that irrigation is still an uncommon practice in this general area, before proposing that a deep water table is detrimental for crop growth. This is a critical information for understanding why crops in this area would depend on natural soil moisture. As it is now, this is only clarified towards the end of the discussion (line 459).

5) Although a shallow water table can be beneficial for vegetation, as thoughtfully discussed in the manuscript, waterlogging also plays an important role in regulating vegetation function and distribution by causing anoxia in the rooting zone (e.g. Rossato et al (2012) for savannas, among several others). Was this considered when classifying the pixels into the two categories? Does this occur in the study area?

6) Why were savannas included in the analysis? Very little is discussed about their characteristics, functioning and why they were of interest to this study. In Figure 2 savannas are lumped with croplands as "other vegetation" (line 229) and hypothesized to have shallow roots, while in reality savanna species can grow roots as deep as or even deeper than forests (Canadell et al, 1996). Besides that, waterlogging is an important driver of distribution and function in Brazilian savannas and therefore special attention should be payed to pixels in the "Shallow WTD" category (as said before in item 5), as they might encompass this condition that is highly detrimental for vegetation.

References

Miguez-Macho G, Fan Y (2012) The role of groundwater in the Amazon water cycle: 2. Influence on seasonal soil moisture and evapotranspiration. J Geophys Res Atmos 117:D15114.

Rossatto, D.R., Silva, L.C.R., Villalobos‐Vega, R., Sternberg, L.S.L. & Franco, A.C. (2012) Depth of water uptake in woody plants relates to groundwater level and vegetation structure along a topographic gradient in a neotropical savanna. Environmental and Experimental Botany 77: 259–266.

Canadell J, et al. (1996) Maximum rooting depth of vegetation types at the global scale. Oecologia 108:583–595.
* * *

---

## Author Comment (AC1) · 10 May 2019

Reviewer 1: O'Conner et al present an interesting and novel study on the effect of ground water depth (GWT) on evapotranspiration (ET), land surface temperature (LST) and the enhanced vegetation index (EVI). More precisely they study if the effect of GWT on ET, LST and EVI differs between vegetation types and season (with respect to rainfall). The authors find a strong difference in ET, LST and EVI between crop- and forest areas. Furthermore for crops they find a higher ET and lower LST in areas with shallower water table depth during the dry season transition.

The topic of the study is interesting and, as far as I know, they are the first to include the effect of GWT on ET at this scale. The set-up of the study (three land cover types,

combined with two WTD classes) is easy to understand and effective. Figure 2 provides a good overview this set-up.

Reply: Thank you for your assessment

I have a few questions for the authors regarding the methodology (the used data sets and different choices made). Afterwards I wrote some general remarks on the content of the paper, followed by some suggestions with respect to the structure of the paper.

Methodology Reviewer 1: Three land cover types are studied: forest, savanna and cropland. The motivation for the inclusion of forest and cropland is clear: both are very different in structure and effect on the moisture recycling system. Also with respect to deforestation, these two land cover classes are a logical choice to study. The motivation for the inclusion of savanna is however not clear to me. What are the characteristics of the Cerrado savanna with respect to the water balance and what can be expected for this ecosystem? To make this more clear, I suggest to add root characteristics of savanna to figure 2.

Reply: We decided to include the savanna land cover class as it is under even greater pressures than forest in terms of land use change; recently cerrado has experienced twice the deforestation rates of forest (Zalles et al. 2019). The savanna system in the Amazon is a very interesting land cover class, as both through land cover or climatic change it hypothesized to represent a new stable system (see for instance Hirota et al. 2011, Staal et al. 2015). We agree with your suggestion to include savanna in Figure 2, thank you. In addition to the updated image we will include more detail on the mixed composition of the cerrado with both tree and grass layers and that this is important both due to differences in leaf area for interception evaporation and transpiration and that the mixed rooting depth of savanna can facilitate hydraulic redistribution (Miguez-Macho & Fan., 2012), where the shallow rooted grasses benefit from the groundwater through the uptake of the deeply rooted trees. We also will add savanna's typical precipitation range (700 to 2000 mm/year), which differs from that of forest (1000-2500

mm/year).

Reviewer 1: Three different time frames are studied: mean annual values, the dry season transition (DST) period and wet season transition (WST) period. The DST and WST are discussed in paragraph 2.3.2. Nevertheless it remains unclear to me why the WST and DST periods are selected, instead of the more extreme dry (and wet) season. Is it related to the planting and harvesting season of the crops?

Reply: Thank you for your comment. Land use land cover change in the arc of deforestation is correlated with a lengthening of the dry season (Dubreuil et al., 2012) and that the onset of the wet season has been related to forest evapotranspiration (Wright et al., 2017). The dry season length is a major concern in the arc of deforestation as future projections indicate drier and longer periods of water stress. We felt that vegetation with deeper rooting depth and/or shallow water table depth increase access to soil moisture effectively delaying the negative effects of the dry season transition and shortening the dry season. While if deeper roots or a shallow water table depth increases access to soil moisture towards the end of the dry season this may increase evapotranspiration and vegetation growth during the wet season transition. Therefore we choose these two periods for further investigation.

We will address this more clearly in the new manuscript why we feel the dry and wet season transition are of special interest.

Reviewer 1:The authors selected the MOD16A2 data product to derive ET and briefly present the product as well as why this product is selected (one of the best available datasets, high spatial and temporal resolution, it is widely used). Also the authors describe that the remote sensing data has "obvious limitations" (L475). I have some concerns regarding this dataset and would like the authors to elaborate a bit on the characteristics and main limitations of using this MODIS data product in their study. Several studies validated the product (E.g. Velpuri et al., 2013) or wrote that especially for tropical sites across the amazon basin, the MODIS ET remains challenging (e.g.

the recently published paper Xu et al., 2019).

Reply: We thank the reviewer for this comment and have added a couple sentences describing the potential and the limitations of the ET product. Xu et al (2019) propose a new method to estimate ET that is better fit to the flux tower data. However, this method has yet to be applied to a remote sensing time series data, and this is beyond the scope of our study. Thus we have added the following in our methods section: "While MODIS ET product is known to be underperforming at fine temporal resolutions and newer novel methods show promising results at nine flux sites across the Amazon (Xu et al. 2019), we believe that the application of the new method for our question on the influence of WTD and our time series analysis was beyond the scope of this study. This is also the reason why we chose to also analyse the effects of WTD on satellite retrieved EVI and LST.."

Reviewer 1: At the studied scale, the modelled water table depth classes are mainly based on the topography of the landscape. Are the MODIS products unbiased for this topography? E.g. is the LST corrected for topography and are the meteorological data required for MODIS ET calculation independent of topography?

Reply: Thank you for your comment. We have now provided more information on the effects of topography on both the ET and LST products. Both account for topography to some extent, explicitly the LST v5 dataset includes issues arising from topography in the quality assessment and in the ET product it is implicit in the meteorological data. We have added this information and it now reads "Each of the three MODIS products used has detailed quality control products allowing low quality pixels to be excluded from the analysis. This removing much of the concern regarding cloud cover or topography.".

Reviewer 1: A few smaller points that unclear to me are: o Are the start and end of the DST and WST calculated for each TRMM pixel? Reply: We thank the reviewer for this question, DST and WST were calculated based on average TRMM across

our study area and not per pixel. We will clarify this in the manuscript o How many (cloudless) remote sensing data points are available? And is this enough to present the results (LST or ET values) with three decimal digits (e.g. L317, 329)? Reply: Thank you for your question. Each of the MODIS datasets contains a quality assessment. Pixels obscured by cloud cover were excluded from the analysis. The percentage of pixels affected by cloud cover is highly correlated with rainfall and therefore impacts our analysis differently. As our number of samples is very large, 1000 pixels per land cover class per timestep even when cloud cover is high we still have a large enough sample size for statistical analysis. Mean number of pixels affected by cloud cover is 40%, during the wet season it reaches as high as 90% while inversely for the dry season < 10% of pixels were affected. The dry season transition had low cloud cover overall with mean cloud cover of 15 - 20 %. On the other hand the wet season transition has relatively high cloud cover with an average of 65 - 70 %.

More information on cloud cover and the number of effective pixels will be included in the methods section 2.3 sampling design. In the discussion section we will discuss how cloud cover may increase uncertainty in our analysis .

General remarks and questions Reviewer 1: The results show that for cropland, EVI is higher for areas with a shallow WTD (paragraph 3.3 / L391). From the supplementary figures, it seems that deep WTD areas lag behind shallow WTD croplands. Is this due to water conditions only, or could this be an effect of a different cropping regime? Do farmers adapt the species and timing of agricultural practices to the local conditions, e.g. length of the dry season?

Reply: We agree with the reviewer comment that the difference seen in crop lands may not be solely driven by water availability. Unfortunately for this study we did not know the details on the local cropping regimes of farmers. It is possible that differences occur between farms and cropping is "optimised" for the local conditions. Sowing of soybean occurs during the wet season transition and can vary between September at October (Gusso et al., 2014). Therefore it is possible that crops in areas with higher water

availability will be sown earlier. Even so, we believe that this indicates that shallow WTD may be better locations for agricultural activity because then crops are not limited in their growth by having to first develop a deep root system. Most crops in the region have shallow root depths (< 2 m – (Setiyono et al., 2008)). We will address this point in the discussion.

Reviewer 1: L184 "This MODIS product ... is correlated to photosynthe­sis/evapotranspiration" (Sims et al., 2006). Please adjust this sentence, or add a refer­ence (Sims et al., 2006 did not study evaporative fluxes).

Reply: Thank you for the comment, we have replaced this reference with Mu et al., 2011.

Reviewer 1: Caption figure 2: "while other vegetation has a lower maximum rooting depth". By other, do you mean crops and/or savanna? What is the rooting depth of savanna trees?

Reply: Thank you for your comment and we agree it was not clear. By other we mean both crops and savanna, with crops having a maximum rooting depth of 2 m and sa­vanna having a maximum rooting depth > 10 m (Canadell et al., 1996). We have added this information to the manuscript.

Reviewer 1: A few lines are unclear to me: o L196 "Further, this choice avoids poten­tial circularity in using land cover classification to detect an effect on a parameter that uses land cover classification to produce its modelled value" Reply: Thank you for your comment. Our main point was that there are advantages in using the same land cover classification maps that are used in the other MODIS products to avoid effects of land cover classification errors across land cover mapping products. We have changed the text and it now reads: " Further, we used the same land cover classification map as is used for the MODIS ET product to avoid effects of land cover classification errors from different maps." o L395 "if this extra warming above the canopy is caused by a change in ET, then better estimates of ET should be possible, however, this is not trivial" Reply:

Thank you for pointed this out. We agree that this line is unclear decided to remove it from the current version of the manuscript o L400 "therefore, the modelled data was expected to underperform, making the differences we found for the dry season even more important" Reply: We agree and have changed this sentence to "Therefore the differences we found for the dry season transition may be under estimated in the MODIS ET values."

Structure and writing

Reviewer 1: I recommend to check the manuscript for spelling, punctuation and sentence structure. Below I give some suggestions that the authors could consider. Reply: thank you, we took those into consideration.

Reviewer 1: I recommend to more clearly differentiate between introduction, methods, results, discussion and conclusion. For example avoid hypothesis-like sentences in the methods section ("We expected that", L178), avoid discussion-like sentences in the results section (e.g. "as hypothesised", L316) and do not add new information to the conclusion.

Reply: We thank you for this comment, and we will carefully go through the manuscript and will remove the more discussion liked phrases from results and methods.

Reviewer 1: Personally I read the lines 490-502 like a discussion, instead of as the conclusion.

Reply: We will move some of the these sentences to the discussion so that they fit the overall text better and give more precise previously introduced conclusions from our paper.

Reviewer 1: Furthermore, I recommend to group the hypotheses in one paragraph and align these hypotheses with the discussion and / or conclusion, to guide the readers through the presented story. Reply: While this is a good suggestion, we also think that this would make a paragraph with all the hypothesis very lengthy. Instead we opted to

divide it into two themes of hypothesis: The relationship of rooting depth and WTD and the resulting effect on vegetation seasonally. We also aligned the discussion along this new structure.

Reviewer 1: From the introduction I deducted four hypotheses and some of them are explicitly discussed, while one is not mentioned in the discussion. Reply: Thank you for pointing this out. We have now included in the discussion comments regarding the delay of the dry season.

Reviewer 1: Also, some new (parts of) hypothesis are introduced in the discussion, which were not introduced earlier. E.g. 1. L102: "We hypothesise that areas of shallow WTD allow vegetation to access soil moisture, with both shallow and deep rooted vegetation potentially facilitating vegetation productivity and higher ET when compared to areas of deep WTD." o L369: "In this study, we tested the hypothesis that areas of shallow WTD would have higher ET when compared to areas of deep WTD, primarily in shallow rooted crop vegetation → Last part of this hypothesis is not mentioned in the introduction. Reply: We agree with your comment and have now included it in the introduction, and it reads as follows "We hypothesize that areas of shallow water table depth (WTD) allow shallow rooted vegetation to access soil moisture, potentially facilitating vegetation productivity and higher evapotranspiration when compared to areas of deep WTD." Reviewer 1: 2. L116: "In areas of shallow WTD, the saturated zone is closer to the root zone of the vegetation. In these locations we, therefore, expect vegetation to be buffered against the reduction in rainfall during the dry season transition and experience drought conditions later, thus delaying the effect of the dry season". → This one is not (directly) referred to in the discussion

Reply: We have now included a sentence in the discussion regarding this prediction and reference the supplemental figures showing the seasonality of the MODIS products and highlight the difference in timing.

Reviewer 1: 3. L428: "As forests has been shown to maintain ET throughout the seasons as its deep roots access deeper groundwater, we hypothesised that no change should be observed in ET, LST and EVI." → I didn't find this hypothesis in the introduction.

Reply: thank you for pointing this out. We have added it to the introduction, and it now reads: "The influence of WTD should be not be visible for deep rooted vegetation like forest and some savanna species."

Reviewer 1: Some spelling related suggestions: • L51 "changes (reduction / decline) in evapotranspiration reduce the available atmospheric moisture". Reply: We have changed it as suggested to "a decline in evapotranspiration reduces" • L57 "forests can maintain a high rate of evapotranspiration during the dry season, they are not affected by low rainfall". Reply: We have changed it as suggested to "Therefore, forest evapotranspiration remains high throughout the year, unaffected by periods of low rainfall" • L92 "agricultural vegetation . . . experiences high seasonality during the dry season unseen in forest vegetation". Seasonality in what? Reply: We have changed it as suggested to "Crops in the Amazon arc of deforestation are impacted by the high seasonality in rainfall affecting vegetation growth unseen in forest vegetation" • L130 "annual average temperatures ranging between 22 – 26 ⁰C". Are 22 and 26 monthly mean temperatures? Reply: These values refer to monthly means and we have included that in the text. • L167 "the MODIS ET products were previously tested . . . more accurate over longer temporal scales and larger areas". By this do you mean more accurate than shorter time/spatial scales? Reply: We have added clarification on what we meant by longer temporal scales and larger areas - "The MODIS ET product was previously tested over the Amazon by comparing its outputs with eddy covariance tower data showing that the ET modelled with MODIS data is more accurate over longer temporal scales (monthly timesteps) and larger spatial extents (e.g. drainage basin)"

• L180 "a 16 day repeated observation". Reply: We have changed it as suggested to "a frequency of 16 days"

• L210 "and found good agreement at/for shallower WTD however,". Reply: We have changed it as suggested to "found good agreement for shallower WTD; however,"

• L229 "these roots may penetrate into the soil until the saturated zone in shallow WTD; however, do not penetrate further in deep WTD". Reply: We have changed it as suggested to "These roots may infiltrate soil until the saturated zone in shallow WTD; however, they cannot penetrate the saturated zone in deep WTD."

• L240 "three primary time periods". Reply: We have changed it as suggested to "three primary time periods"

• L265 "we used an average value over these transition periods" (value of what?). Reply: We have changed it as suggested to "We used an average of each remote sensing product over these transition"

• L287 "a year was considered statistically significant". E.g. "for one year, the difference in .. was considered statistically significant". Reply: We have changed it as suggested to "For one year, the difference in ET,EVI or LST was considered statistically significant when. . ."

• L302 "3.967 $\pm$ 0.0.09". Reply: We have changed it as suggested to "0.09"

• E.g. L371/L379 "since crop experiences". I recommend to use for example "crop species" or "a crop" or "cropland". Reply: We have changed it as suggested to "a crop experiences"

• L377 "indicate that local conditions can be much warmer in deep WTD areas". Reply: We have changed it as suggested to "indicate that LST in deep WTD areas can reach much higher temperatures than shallow WTD areas."

• L380 "the roots of crop vegetation only penetrates to a maximum of 2 m, in shallow..". Reply: We have changed it to "The maximum rooting depth for most crops in the region is 2 m, in shallow. . ."
• L408 "this could mean that in deep WTD areas temperature could even be .." Reply: We have changed it to "This increase in temperature could be influenced by WTD and land cover change; in shallow WTD areas this may result in a less severe temperature change while in deep WTD it could lead to a greater change in temperature"

• L409 "WTD was not". Reply: We have changed it as suggested

• L444 "the difference in ET was very small, < 1% difference between deep and shallow rooted areas". Reply: We have changed it as suggested to "The difference in ET was very small, less than 1 % difference between deep and shallow WTD areas".

---

## Author Comment (AC2) · 10 May 2019

Reviewer 2 In this paper, O'Connor et al tackle an interesting and very important question in the field of ecohydrology: how does groundwater affect plant functioning? As a community, it is important that we move from the broad, large-scale influences of climate towards focusing on the regional to local scales, where, as shown by several authors, groundwater might be one of the driving forces of ecosystems. This has important implications for our understanding of the response of natural and agricultural systems to climate change, and this study is a timely contribution to this field. I believe, however, that there are some conceptual and methodological issues with this study that should be addressed before publication. Below I offer some comments on the content of the paper and also pose some questions that might help the authors in

further refining it. Specific comments

1) The ultimate focus or "big question" of this study was somewhat unclear to me as I read the paper. In the introduction, a lot of importance is given to large scale problems such as the impact of land use changes on precipitation recycling and the subsequent negative effect on forest cover through a reduction in ET. However, in the conclusions section, the "key messages" are related to agricultural management and forest conservation. I believe the paper would greatly benefit from a clear, defined question that is posed in the beginning of the paper and that guides the discussion and conclusions. Reply: Thank you for your comment. Our main objective is to study the effect of water table depth (WTD) on evapotranspiration (ET) across the different land covers in the Amazon using remote sensing. Indeed there is a strong emphasis on the role of evapotranspiration and precipitation recycling in the introduction, as this is a highly important ecosystem service in the region. During the course of our analysis we recognised that WTD did not have a major impact on ET when compared to the much larger issue of land cover change. Although we did not find support for an effect on the precipitation recycling system we still felt that it was good to frame our study in the larger context. We agree that the introduction does not currently align with the main take home message. In order to strengthen the findings in this paper we will add information to the introduction regarding the agricultural system and local effects of evapotranspiration.

From the climate system point of view, the small differences in ET between shallow and deep WTD observed in the study might not be significant, while from the perspective of sustainable agricultural management and general crop productivity these changes might suggest a more water efficient practice. Perhaps the authors could group their questions with their hypotheses, which currently are somewhat scattered throughout the introduction and methodology sections.

Reply: Thank you for the comment. We agree and have regrouped the hypotheses into two main themes in the introduction which are then traced back in the results,

discussion and conclusion.

2) Although groundwater is the main environmental factor addressed in this study, very little is discussed about it throughout the paper. What is topography like in the region of study? How does the water table field look like in this area? How deep and how shallow can the water table be? What is the meaning of an "equilibrium water table depth"? What are the benefits and the drawbacks of using an equilibrium water table instead of a dynamic product? Is this an area where the water table responds directly to precipitation or is lateral convergence an important process? These are some key questions that directly impact the hypotheses and conclusion of this study, and therefore should be well addressed in the manuscript.

Reply: Thank you for this comment. We recognise that more information is needed to accurately describe the water table characteristics. We will make sure to include information regarding the maximum water table depth with in our area and add to the description of the modelled used that the "equilibrium" water table depth used is a long term average depth. This model was chosen as it was the best available fit for the spatial and temporal scale. As our goal was not to model evapotranspiration or the water system ourselves we did not want to try and separately simulate a dynamic water table depth. Therefore we choose our distinct shallow (< 2 m) and deep (> 10 m) categories as they are robust for our purposes.

3) I don't understand the reasoning behind choosing the wet and dry season transitions as periods of stress for vegetation. The use of a climatic index neglects the important time lag displayed by groundwater (and soil moisture in general) that has been shown to support considerable levels of ET well into the dry season for several places in the Amazon basin (Miguez-Macho and Fan, 2012). In fact, seasonal soil moisture storage maps from Miguez-Macho and Fan (2012) show that, in the top 2 m, October is a more critical month in this general area than the dry season transition (June/July) proposed here by the authors. Is there a specific reason for choosing these periods?
Reply: Thank you very much for this question, it was also somewhat raised by reviewer 1. The choice to use the dry season and wet season transition periods was based on the idea that land use change is leading to a lengthening of the dry season and that the high forest evapotranspiration is integral in initiating the wet season. We therefore wanted to examine how access to soil moisture would effect evapotranspiration during these periods. We agree that the time lag between deep and shallow rooted vegetation is an important aspect and will include this in our results / discussion with reference to the seasonality figures in the supplemental information.

4) The authors should include early in the introduction that irrigation is still an uncommon practice in this general area, before proposing that a deep water table is detrimental for crop growth. This is a critical information for understanding why crops in this area would depend on natural soil moisture. As it is now, this is only clarified towards the end of the discussion (line 459). Reply: We agree that more information is needed in the test about the limited use of irrigation. We will add information in the introduction to introduce the reliance on precipitation and the limited application of irrigation.

5) Although a shallow water table can be beneficial for vegetation, as thoughtfully discussed in the manuscript, waterlogging also plays an important role in regulating vegetation function and distribution by causing anoxia in the rooting zone (e.g. Rossato et al (2012) for savannas, among several others). Was this considered when classifying the pixels into the two categories? Does this occur in the study area?

Reply: Thank you for your comment, which is very valid. We are unaware of waterlogging occurring in forest area, waterlogging is an important driver of distribution and function in Brazilian savannas. Nonetheless, because pixels were selected when they were consistently classified as the same land cover type for 12 consecutive years, which we would expect not to be the case if waterlogging had happened as it would lead to changes in land cover. We added the following to the Methods secion 2.3.2 Data analysis to further clarify: "and vegetation distribution as waterlogging of soils can lead to anoxia in the root zone. Due to the selection of only consistently classified

pixels the influence of water logging can be avoided as over time these areas will fall under different classifications"

6) Why were savannas included in the analysis? Very little is discussed about their characteristics, functioning and why they were of interest to this study. In Figure 2 savannas are lumped with croplands as "other vegetation" (line 229) and hypothesized to have shallow roots, while in reality savanna species can grow roots as deep as or even deeper than forests (Canadell et al, 1996). Besides that, waterlogging is an important driver of distribution and function in Brazilian savannas and therefore special attention should be payed to pixels in the "Shallow WTD" category (as said before in item 5), as they might encompass this condition that is highly detrimental for vegetation.

Reply: Thank you for the comment. We have responded to a similar comment by reviewer 1 above. We have added more information of why including cerrado savanna, we also included it in Figure 2 and the explained that savanna species can grow roots as deep as or even deeper than forests by adding references to rooting depths by land cover type. We will also introduce that the deep roots of tree species in savanna's can increase soil moisture available to the shallow rooted grasses via hydraulic redistribution. In the discussion of the "mixed" results seen in the savanna data we will include the possibility that waterlogging may drive vegetation patterns and distribution of different savanna types. We will also discuss the

---

## Author Response (AR1)

Dear editor,

Thank you very much for this opportunity. We are very grateful for the constructive comments of both reviewers and believe that their recommendations with result in a much improved version of the manuscript. As recommended we have taken special attention to look into the topography of our study area. We examined both elevation and topographic wetness index for bias which could influence the results of our study. Details of which our now included in both the text and the updated supplemental information. Below The reviewers comments are followed by the author reply and changed made in the paper, highlighted in blue. Where relavant line numbers were given in parenthesis.

Kind regards

John O'Connor

**Reviewer 1:** O'Conner et al present an interesting and novel study on the effect of ground water depth (GWT) on evapotranspiration (ET), land surface temperature (LST) and the enhanced vegetation index (EVI). More precisely they study if the effect of GWT on ET, LST and EVI differs between vegetation types and season (with respect to rainfall). The authors find a strong difference in ET, LST and EVI between crop- and forest areas. Furthermore for crops they find a higher ET and lower LST in areas with shallower water table depth during the dry season transition.

The topic of the study is interesting and, as far as I know, they are the first to include the effect of GWT on ET at this scale. The set-up of the study (three land cover types, combined with two WTD classes) is easy to understand and effective. Figure 2 provides a good overview this set-up.

**Reply:** Thank you for your assessment

I have a few questions for the authors regarding the methodology (the used data sets and different choices made). Afterwards I wrote some general remarks on the content of the paper, followed by some suggestions with respect to the structure of the paper.

**Methodology**
**Reviewer 1:** Three land cover types are studied: forest, savanna and cropland. The motivation for the inclusion of forest and cropland is clear: both are very different in structure and effect on the moisture recycling system. Also with respect to deforestation, these two land cover classes are a logical choice to study. The motivation for the inclusion of savanna is however not clear to me. What are the characteristics of the Cerrado savanna with respect to the water balance and what can be expected for this ecosystem? To make this more clear, I suggest to add root characteristics of savanna to figure 2.

**Reply:** We decided to include the savanna land cover class as it is under even greater pressures than forest in terms of land use change; recently cerrado has experienced twice the deforestation rates of forest (Zalles et al. 2019).

We have updated figure 2 to now include the savanna land cover class.

In addition to the updated image we have included savanna more throughout the paper. Introducing how it compares to forest and cropland in terms of rooting depth and LAI (Line 55-56, 62-63).

We also will add savanna's typical precipitation range (700 to 2000 mm/year), which differs from that of forest (1000-2500 mm/year). (Line 145)

**Reviewer 1:** Three different time frames are studied: mean annual values, the dry season transition (DST) period and wet season transition (WST) period. The DST and WST are discussed in paragraph 2.3.2. Nevertheless it remains unclear to me why the WST and DST periods are selected, instead of the more extreme dry (and wet) season. Is it related to the planting and harvesting season of the crops?

**Reply:** Thank you for your comment. Land use land cover change in the arc of deforestation is correlated with a lengthening of the dry season (Dubreuil et al., 2012) in particular results suggest a delay in the wet season. The DST and WST are calculated using Liebmann's anomalous accumulation method. The DST occurs after an already significant decline in rainfall compared with the wet season. We expect that during this period vegetation with access to ground water will experience less stress than vegetation without. Similarly the WST rainfall has already started to increase but vegetation with access to ground water will be able to meet their demand for water easier increasing growth. This gave us an interesting period when access to ground water could make the largest difference. We did not focus on the extreme dry as during this period crop ET was suppressed for both groups probably due to additional factors outside of water availability i.e. heat stress.

In section 2.3.2 data analysis we added our motivation for selecting the WST and the DST.. (Line 316-319)

**Reviewer 1:** The authors selected the MOD16A2 data product to derive ET and briefly present the product as well as why this product is selected (one of the best available datasets, high spatial and temporal resolution, it is widely used). Also the authors describe that the remote sensing data has "obvious limitations" (L475). I have some concerns regarding this dataset and would like the authors to elaborate a bit on the characteristics and main limitations of using this MODIS data product in their study. Several studies validated the product (E.g. Velpuri et al., 2013) or wrote that especially for tropical sites across the amazon basin, the MODIS ET remains challenging (e.g. the recently published paper Xu et al., 2019).

**Reply:** We thank the reviewer for this comment and have added a couple sentences describing the potential and the limitations of the ET product. Xu et al (2019) propose a new method to estimate ET that is better fit to the flux tower data. However, this method has yet to be applied to a remote sensing time series data, and this is beyond the scope of our study.

Thus we have added the following in our methods section: "While MODIS ET product is known to be underperforming at fine temporal resolutions and newer novel methods show promising results at nine flux sites across the Amazon (Xu et al. 2019), we believe that the application of the new method for our question on the

influence of WTD and our time series analysis was beyond the scope of this study. This is also the reason why we chose to also analyse the effects of WTD on satellite retrieved EVI and LST.."

**Reviewer 1:** At the studied scale, the modelled water table depth classes are mainly based on the topography of the landscape. Are the MODIS products unbiased for this topography? E.g. is the LST corrected for topography and are the meteorological data required for MODIS ET calculation independent of topography?

**Reply:** Thank you for your comment. Indeed topography can influence both WTD and ET possibly leading to bias. The MODIS LST limits strong impacts of topography and these pixels are disqualified from our study using the QA band. In order to investigate further about the influence of topography. We looked at elevation (which impact meteorological forcing of the ET model) and calculated topographic wetness per pixel (Uses a combination of slope and contributing area) of our randomly selected pixels. We found no differences in elevation between deep and shallow WTD for forest and savanna. Elevation in crop areas was statistically different in 50% of the randomisations, however, the mean difference in elevation is only 10m so we do not believe that this difference is driving differences seen in ET. We found no statistical differences in topographic wetness index between deep and shallow WTD for any of the land cover classes.

We have added this information to the text (lines 220-231) and also included elevation and topographic index figures in the supplemental information SI.9.

**Reviewer 1:** A few smaller points that unclear to me are:

 o Are the start and end of the DST and WST calculated for each TRMM pixel?

 **Reply:** We thank the reviewer for this question, DST and WST were calculated based on average TRMM across our study area and not per pixel.

 We have clarified this in the manuscript (line 235)

 o How many (cloudless) remote sensing data points are available? And is this enough to present the results (LST or ET values) with three decimal digits (e.g. L317, 329)?

 **Reply:** Thank you for your question. Each of the MODIS datasets contains a quality assessment. Pixels obscured by cloud cover were excluded from the analysis. The percentage of pixels affected by cloud cover is highly correlated with rainfall and therefore impacts our analysis differently. As our number of samples is very large, 1000 pixels per land cover class per timestep even when cloud cover is high we still have a large enough sample size for statistical analysis. Mean number of pixels affected by cloud cover is 40%, during the wet season it reaches as high as 90% while inversely for the dry season < 10% of pixels were affected. The dry season transition had low cloud cover overall with mean cloud cover of 15 - 20 %. On the other hand the wet season transition has relatively high cloud cover with an average of  65 - 70 %.

We have added information on cloud cover (lines 215-219). We have also added a figure to the supplemental information showing the seasonality of pixels impacted by cloud cover (SI.10.1)

**General remarks and questions**

**Reviewer 1:** The results show that for cropland, EVI is higher for areas with a shallow WTD (paragraph 3.3 / L391). From the supplementary figures, it seems that deep WTD areas lag behind shallow WTD croplands. Is this due to water conditions only, or could this be an effect of a different cropping regime? Do farmers adapt the species and timing of agricultural practices to the local conditions, e.g. length of the dry season?

**Reply:** We agree with the comment that the difference seen in crop lands may not be solely driven by water availability. Unfortunately for this study we did not know the details on the local cropping regimes of farmers. It is possible that differences occur between farms and cropping is "optimised" for the local conditions. Sowing of soybean occurs during the wet season transition and can vary between September at October (Gusso et al., 2014). Therefore it is possible that crops in areas with higher water availability will be sown earlier. Even so, we believe that this indicates that shallow WTD may be better locations for agricultural activity because then crops are not limited in their growth by rainfall.
We included this point in the discussion. (lines 528-533)

**Reviewer 1:** L184 "This MODIS product … is correlated to photosynthesis/evapotranspiration" (Sims et al., 2006). Please adjust this sentence, or add a reference (Sims et al., 2006 did not study evaporative fluxes).

**Reply:** Thank you for the comment, we have replaced this reference with Mu et al., 2011.

**Reviewer 1:** Caption figure 2: "while other vegetation has a lower maximum rooting depth". By other, do you mean crops and/or savanna? What is the rooting depth of savanna trees?

**Reply:** Thank you for your comment and we agree it was not clear. By other we mean both crops and savanna, with crops having a maximum rooting depth of 2 m and savanna having a maximum rooting depth > 10 m (Canadell et al., 1996).
We have adjusted figure 2 as well as the caption in the manuscript.

**Reviewer 1:** A few lines are unclear to me:
  o L196 "Further, this choice avoids potential circularity in using land cover classification to detect an effect on a parameter that uses land cover classification to produce its modelled value"
**Reply:** Thank you for your comment. Our main point was that there are advantages in using the same land cover classification maps that are used in the other MODIS products to avoid effects of land cover classification errors across land cover mapping products.
We have changed the text to make this more clear (lines 212-214): " Further, we used MODIS land cover as it is the same land cover classification map as used for the MODIS ET product to avoid effects of land cover classification errors from different maps."
  o L395 "if this extra warming above the canopy is caused by a change in ET, then better estimates of ET should be possible, however, this is not trivial"

**Reply:** Thank you for pointed this out. We agree that this line is unclear decided to remove it from the current version of the manuscript

> o L400 "therefore, the modelled data was expected to underperform, making the differences we found for the dry season even more important"

**Reply:** We agree and have changed this sentence to (line 457-458): "Therefore the differences we found for the dry season transition may be under estimated in the MODIS ET values."

**Structure and writing**

**Reviewer 1:** I recommend to check the manuscript for spelling, punctuation and sentence structure. Below I give some suggestions that the authors could consider.

**Reply:** Thank you, we took those into consideration.

**Reviewer 1:** I recommend to more clearly differentiate between introduction, methods, results, discussion and conclusion.

For example avoid hypothesis-like sentences in the methods section ("We expected that", L178), avoid discussion-like sentences in the results section (e.g. "as hypothesised", L316) and do not add new information to the conclusion.

**Reply:** We thank you for this comment, and we have edited the manuscript and removed these phrases from results and methods.

**Reviewer 1:** Personally I read the lines 490-502 like a discussion, instead of as the conclusion.

**Reply:** Thank you for your feedback. We agree and have move some of the previous conclusion section to the discussion and rewritten our conclusion.

**Reviewer 1:** Furthermore, I recommend to group the hypotheses in one paragraph and align these hypotheses with the discussion and / or conclusion, to guide the readers through the presented story.

Reply: While this is a good suggestion, we also think that this would make a paragraph with all the hypothesis very lengthy. Instead we opted to divide it into two themes of hypothesis: The influence of WTD on crop (shallow rooted vegetation) and the resulting effect on vegetation seasonally. The influence (or lack thereof) on forest and savanna (deep rooted vegetation). We also aligned the discussion along this new structure.

**Reviewer 1:** From the introduction I deducted four hypotheses and some of them are explicitly discussed, while one is not mentioned in the discussion.

**Reply:** Thank you for pointing this out. We have restructured and improved the connection between hypothesis and discussion

**Reviewer 1:** Also, some new (parts of) hypothesis are introduced in the discussion, which were not introduced earlier. E.g.

1. L102: "We hypothesise that areas of shallow WTD allow vegetation to access soil moisture, with both shallow and deep rooted vegetation potentially facilitating vegetation productivity and higher ET when compared to areas of deep WTD."

   o L369: "In this study, we tested the hypothesis that areas of shallow WTD would have higher ET when compared to areas of deep WTD, primarily in shallow rooted crop vegetation ◊ Last part of this hypothesis is not mentioned in the introduction.

   **Reply:** We agree with your comment and have now edited and improved the connection between introduction and discussion. Line 124-125 addressing the hypothesis that shallow rooted vegetation will show the greatest differences between deep and shallow WTD

**Reviewer 1:** 2. L116: "In areas of shallow WTD, the saturated zone is closer to the root zone of the vegetation. In these locations we, therefore, expect vegetation to be buffered against the reduction in rainfall during the dry season transition and experience drought conditions later, thus delaying the effect of the dry season". ◊ This one is not (directly) referred to in the discussion

**Reply:** We have now included a sentence in the discussion regarding this prediction and reference the supplemental figures showing the seasonality of the MODIS products and highlight the difference in timing. Line(449-451)

**Reviewer 1:** 3. L428: "As forests has been shown to maintain ET throughout the seasons as its deep roots access deeper groundwater, we hypothesised that no change should be observed in ET, LST and EVI." ◊ I didn't find this hypothesis in the introduction.

**Reply:** Thank you for pointing this out. We have added it to the introduction, and it now reads: "As reported in other studies the influence of WTD should not be visible for deep rooted vegetation (Nepstad et al., 1994) like forest and some savanna species."

**Reviewer 1:** Some spelling related suggestions:

• L51 "changes (reduction / decline) in evapotranspiration reduce the available atmospheric moisture".
**Reply:** We have changed it as suggested to "a decline in evapotranspiration reduces"

• L57 "forests can maintain a high rate of evapotranspiration during the dry season, they are not affected by low rainfall".
**Reply:** We have changed it as suggested to "Therefore, forest evapotranspiration remains high throughout the year, unaffected by periods of low rainfall" (line 61)

• L92 "agricultural vegetation … experiences high seasonality during the dry season unseen in forest vegetation". Seasonality in what?

**Reply:** We have changed it as suggested to "Crops in the Amazon arc of deforestation are predominantly rainfed and as such impacted by the high seasonality in rainfall unseen in forest vegetation" (Line 97-98)

• L130 "annual average temperatures ranging between 22 – 26 ºC". Are 22 and 26 monthly mean temperatures?

**Reply:** These values refer to monthly means and we have included that in the text.

• L167 "the MODIS ET products were previously tested … more accurate over longer temporal scales and larger areas". By this do you mean more accurate than shorter time/spatial scales?

**Reply:** We have added clarification on what we meant by longer temporal scales and larger areas - "The MODIS ET product was previously tested over the Amazon by comparing its outputs with eddy covariance tower data showing that the ET modelled with MODIS data is more accurate over longer temporal scales (monthly timesteps) and larger spatial extents (e.g. drainage basin)" (lines 181-184)

• L180 "a 16 day repeated observation".

**Reply:** We have changed it as suggested to "a frequency of 16 days" (line 198)

• L210 "and found good agreement at/for shallower WTD however,".

**Reply:** We have changed it as suggested to "found good agreement for shallower WTD; however," (lines 246-247)

• L229 "these roots may penetrate into the soil until the saturated zone in shallow WTD; however, do not penetrate further in deep WTD".

Reply: We have changed figure 2's caption to "Figure 2: Diagram showing that forest (A) root depth can reach until the saturated zone in both shallow (< 2 m) and deep (> 8 m) WTD, savanna (B) has a mixed rooting depth with only tree roots reaching deep WTD and crop (C) vegetation have a low maximum rooting depth (crops having a maximum rooting depth of 2 m and savanna having a maximum rooting depth > 10 m (Canadell et al., 1996). Shallow roots can reach the saturated zone in shallow WTD (< 2 m); however, they cannot reach the saturated zone in deep WTD (> 8 m)."

• L240 "three primary time periods".

**Reply:** We have changed it as suggested to "three primary time periods" (Line 285)

• L265 "we used an average value over these transition periods" (value of what?).

**Reply:** We have changed it as suggested to "We used an average of each remote sensing product over these transition" (Line 302)

• L287 "a year was considered statistically significant". E.g. "for one year, the difference in .. was considered statistically significant".

**Reply:** We have changed it as suggested to "For one year, the difference in ET,EVI or LST was considered statistically significant when…" (Line 336-337)

• L302 "3.967 ± 0.0.09".

**Reply:** We have corrected it to "0.09" (Line 351)

• E.g. L371/L379 "since crop experiences". I recommend to use for example "crop species" or "a crop" or "cropland".

**Reply:** We have changed it as suggested to "since crops experience" (line 429)

• L377 "indicate that local conditions can be much warmer in deep WTD areas".

**Reply:** We have changed it as suggested to "indicate that LST in deep WTD areas can reach much higher temperatures than shallow WTD areas." (Line 434)

• L380 "the roots of crop vegetation only penetrates to a maximum of 2 m, in shallow..".

Reply: We have changed it to "The maximum rooting depth for most crops in the region is 2 m, in shallow…" (Line 437-438)

• L408 "this could mean that in deep WTD areas temperature could even be .."

**Reply:** We have changed it to "This increase in temperature could be influenced by WTD and land cover change; in  shallow WTD areas this may result in a less severe temperature change while in deep WTD it could lead to a greater change in temperature" (Lines 465-467)

• L409 "WTD was not".

**Reply:** We have changed it as suggested (Line 467)

• L444 "the difference in ET was very small, < 1% difference between deep and shallow rooted areas".

**Reply:** We have changed it as suggested to "The difference in ET was very small, less than 1 % difference between deep and shallow WTD areas".  (Lines 502-503)

**Reviewer 2**

In this paper, O'Connor et al tackle an interesting and very important question in the field of ecohydrology: how does groundwater affect plant functioning? As a community, it is important that we move from the broad, large-scale influences of climate towards focusing on the regional to local scales, where, as shown by several authors, groundwater might be one of the driving forces of ecosystems. This has important implications for our understanding of the response of natural and agricultural systems to climate change, and this study is a timely contribution to this field. I believe, however, that there are some conceptual and methodological issues with this study that should be addressed before publication. Below I offer some comments on the content of the paper and also pose some questions that might help the authors in further refining it.

Thank you very much for your assessment of the paper and interest in the topic. Your comments were very helpful in improving this paper

Specific comments

**1)** The ultimate focus or "big question" of this study was somewhat unclear to me as I read the paper. In the introduction, a lot of importance is given to large scale problems such as the impact of land use changes on precipitation recycling and the subsequent negative effect on forest cover through a reduction in ET. However, in the conclusions section, the "key messages" are related to agricultural management and forest conservation. I believe the paper would greatly benefit from a clear, defined question that is posed in the beginning of the paper and that guides the discussion and conclusions.

**Reply:** Thank you for your comment. Our main objective is to study the effect of water table depth (WTD) on evapotranspiration (ET) across the different land covers in the Amazon using remote sensing. Indeed there is a strong emphasis on the role of evapotranspiration and precipitation recycling in the introduction, as this is a highly important ecosystem service in the region. During the course of our analysis we recognised that WTD did not have a major impact on ET when compared to the much larger issue of land cover change. Although we did not find support for an effect on the precipitation recycling system we still felt that it was good to frame our study in the larger context.

We agree that the introduction does not currently align with the main take home message. In order to strengthen the findings in this paper we have added information to the introduction regarding the agricultural system and local effects of evapotranspiration. We have also changed our conclusion section in order to strengthen the position of our paper.

From the climate system point of view, the small differences in ET between shallow and deep WTD observed in the study might not be significant, while from the perspective of sustainable agricultural management and general crop productivity these changes might suggest a more water efficient practice. Perhaps the authors could group their questions with their hypotheses, which currently are somewhat scattered throughout the introduction and methodology sections.

**Reply:** Thank you for the comment. We agree and have regrouped the hypotheses into two main themes in the introduction which are then traced back in the results, discussion and conclusion. The influence of WTD on shallow rooted crop vegetation and the resulting effect on seasonally. The ability of deep roots (forest and savanna) to evenly access deep and shallow WTD.

**2)** Although groundwater is the main environmental factor addressed in this study, very little is discussed about it throughout the paper. What is topography like in the region of study? How does the water table field look like in this area? How deep and how shallow can the water table be? What is the meaning of an "equilibrium water table depth"? What are the benefits and the drawbacks of using an equilibrium water table instead of a dynamic product? Is this an area where the water table responds directly to precipitation or is lateral convergence an important process? These are some key questions that directly impact the hypotheses and conclusion of this study, and therefore should be well addressed in the manuscript.

**Reply:** Thank you for this comment. We recognise that more information is needed to accurately describe the water table characteristics. We have included information regarding the water table depth within our study area (line 148, SI.8) and add to the description of the modelled used that the "equilibrium" water table depth used is a long term mean depth (line 238). This model was chosen as it was the best available fit for the spatial scale. As our goal was not to model evapotranspiration or the water system ourselves we did not want to try and separately simulate a dynamic water table depth. Therefore we choose our distinct shallow (< 2 m) and deep (> 10 m) categories as they are robust for our purposes. We have also added information about the elevation of our study area (lines 146-147, SI.9) and discussed the influence topography can have on our data (lines 225-231).

**3)** I don't understand the reasoning behind choosing the wet and dry season transitions as periods of stress for vegetation. The use of a climatic index neglects the important time lag displayed by groundwater (and soil moisture in general) that has been shown to support considerable levels of ET well into the dry season for several places in the Amazon basin (Miguez-Macho and Fan, 2012). In fact, seasonal soil moisture storage maps from Miguez-Macho and Fan (2012) show that, in the top 2 m, October is a more critical month in this general area than the dry season transition (June/July) proposed here by the authors. Is there a specific reason for choosing these periods?

**Reply:** Thank you very much for this question, it was also somewhat raised by reviewer 1. The choice to use the dry season and wet season transition periods was based on the idea that land use change is leading to a lengthening of the dry season. As crop ET is suppressed at the height of the dry season we therefore wanted to examine how access to soil moisture would effect evapotranspiration during these periods. The calculation used in this study sets DST and WST after a change in precipitation has already occurred (from the highest point of the wet or dry season). This allows us to investigate the early effects of a reduction or increase in rainfall.
We have made an effort to express our motivation for using these periods in the updated version of the paper. (Lines 313 – 318)

**4)** The authors should include early in the introduction that irrigation is still an uncommon practice in this general area, before proposing that a deep water table is detrimental for crop growth. This is a critical information for

understanding why crops in this area would depend on natural soil moisture. As it is now, this is only clarified towards the end of the discussion (line 459).

Reply: We agree that more information is needed in the test about the limited use of irrigation.
We have added information in the introduction to introduce the reliance on precipitation and the limited application of irrigation. (Lines 97-98, 113-118)

5) Although a shallow water table can be beneficial for vegetation, as thoughtfully discussed in the manuscript, waterlogging also plays an important role in regulating vegetation function and distribution by causing anoxia in the rooting zone (e.g. Rossato et al (2012) for savannas, among several others). Was this considered when classifying the pixels into the two categories? Does this occur in the study area?

Reply: Thank you for your comment, which is very valid. We are unaware of waterlogging occurring in forest area, waterlogging is an important driver of distribution and function in Brazilian savannas. Nonetheless, because pixels were selected when they were consistently classified as the same land cover type for 12 consecutive years, which we would expect not to be the case if waterlogging had happened as it would lead to changes in land cover. We added the following to the Methods secion 2.3.2 Data analysis to further clarify: "and vegetation distribution as waterlogging of soils can lead to anoxia in the root zone. Due to the selection of only consistently classified pixels the influence of water logging can be avoided as over time these areas will fall under different classifications" (Lines 279-283) In the discussion about savannas we included the possibility that waterlogging may drive vegetation patterns and distribution (Lines 503-507)

6) Why were savannas included in the analysis? Very little is discussed about their characteristics, functioning and why they were of interest to this study.
In Figure 2 savannas are lumped with croplands as "other vegetation" (line 229) and hypothesized to have shallow roots, while in reality savanna species can grow roots as deep as or even deeper than forests (Canadell et al, 1996). Besides that, waterlogging is an important driver of distribution and function in Brazilian savannas and therefore special attention should be payed to pixels in the "Shallow WTD" category (as said before in item 5), as they might encompass this condition that is highly detrimental for vegetation.

Reply: Thank you for the comment. We have responded to a similar comment by reviewer 1 above. We have added more information of why including cerrado savanna, we also included savanna as a separate land cover class in Figure 2. Throughout the paper we reference the deep rooting depth of savanna trees and how this leads to access to deep ground water similar to forests.

---

## Referee Report (RR1)

With pleasure I read the revised manuscript by O'Conner et al on the effect of ground water depth on evapotranspiration, land surface temperature and vegetation growth. I was also one of the reviewers of the first version of the manuscript. The authors well answered to questions raised by both reviewers and incorporated the suggestions.

Savanna is added to the manuscript as a distinct third land cover class, for example by adding it in fig. 2 and by discussing its characteristics and interest. This, together with the clear connection between the introduction (specifically paragraph 4) and discussion, greatly improved the structure of the paper. The methods section is clear and complete and the authors addressed all concerns of the reviewers in their revised manuscript.

I have two small questions remaining and I have a few language related suggestions that the authors might want to look into. This is not a complete list, but just a few things that I came across when reading the paper.

L503: "water logging of soils has been shown to be an important factor in determining vegetation distribution". Is this statement based on literature?

L470: "we also expected that the influence of WTD would be most important during the wet season transition". This hypothesis is not phrased in the introduction. Do you hypothesise the influence of WTD to be more important during the WST compared to the DST? If so, why?

- I suggest to define the abbreviations once and use them consistently throughout the document, e.g. DST and WST.
- L55 "has lower leaf surface area"
- L56 "and agricultural vegetation usually ha**s**"
- L102 "agricultural crops are known"
- L107 "thus access to soil moisture is an important limiting factor for photosynthesis and transpiration". Do you mean limited access to soil moisture?
- L127 "we expect to find"
- L146 "with a maximum of 700 **m and** a minimum of 100 m"
- L194 "Despite low albedo (..) and high net radiation". What do you mean by despite?
- L200 "as **EVI** is less sensitive"
- L217 "due to our large sample size, we still **have** enough"
- L220 / L232: both start with finally
- L231 "SI.9**.4**"
- L234 "which was then applied to an accumulation to calculate" This sentence is unclear to me.
- L247 "the model overestimated deep WTD" / L458 "may be underestimated"
- L251 "these depths were selected as they represent rooting depth"
- L270 "crop vegetation **has** a low"
- L310 "The WST and DST periods were selected .."
- L318 "shallow rooted vegetation"
- L376 "effect of ground water depth **on** Land Surface Temperature"
- L401 "between deep and shallow **WTD**"
- L412 with between, do you mean among?

- L425 "we expect to see the largest influence of WTD in crop vegetation".
- L428 "was not different between crop **areas** in .."
- L430 replace "a high / a low" with for example a maximum / a minimum"
- L440 "and therefore evaporative cooling **could explain** the cooler"
- L476 "higher in shallow than deep WTD **areas**"
- L475 "during the WST EVI is increasing / increases faster"
- L531 "can vary between September at October". Do you mean September and October?
- L561 the sentence "on a local scale show signs that .." is unclear to me.

---

## Author Response (AR2)

Dear editor,

Thank you very much for the time and attention afforded to us. We were very happy to read the assessment of the reviews and hope that with this version of the manuscript to have addressed all their remaining comments. Please find below the reviewer's comments with our reply and changes made in the paper (highlighted in blue).

Kind regards

John O'Connor

With pleasure I read the revised manuscript by O'Conner et al on the effect of ground water depth on evapotranspiration, land surface temperature and vegetation growth. I was also one of the reviewers of the first version of the manuscript. The authors well answered to questions raised by both reviewers and incorporated the suggestions. Savanna is added to the manuscript as a distinct third land cover class, for example by adding it in fig. 2 and by discussing its characteristics and interest. This, together with the clear connection between the introduction (specifically paragraph 4) and discussion, greatly improved the structure of the paper. The methods section is clear and complete and the authors addressed all concerns of the reviewers in their revised manuscript. I have two small questions remaining and I have a few language related suggestions that the authors might want to look into. This is not a complete list, but just a few things that I came across when reading the paper.

Thank you for your kind assessment and helpful comments

L503: "water logging of soils has been shown to be an important factor in determining vegetation distribution". Is this statement based on literature?

Thank you for your comment. We have added two references related to the influence of water table depth on vegetation distribution. Ridolfi et al. (2006) and Rossatto et al. (2012) (Originally suggested in the previous round of review).

L470: "we also expected that the influence of WTD would be most important during the wet season transition". This hypothesis is not phrased in the introduction. Do you hypothesise the influence of WTD to be more important during the WST compared to the DST? If so, why?

Thank you for your question and sorry for the confusion. It was not our intention to imply one period was more important than the other but that the WST and DST were both important periods in the year. For this reason we have removed the word "most" so the text now reads "we also expected that the influence of WTD would be important during the WST".

• I suggest to define the abbreviations once and use them consistently throughout the document, e.g. DST and WST.

Changed: Dry season transition and Wet season transition have now been changed to the abbreviation DST and WST throughout the manuscript starting from the 2.3.2

- L55 "has lower leaf surface area"

    Changed as suggested

- L56 "and agricultural vegetation usually has"

    Changed as suggested

- L102 "agricultural crops are known"

    Changed as suggested

- L107 "thus access to soil moisture is an important limiting factor for photosynthesis and transpiration". Do you mean limited access to soil moisture?

    Changed as suggested to limited access to soil moisture

- L127 "we expect to find"

    Changed as suggested

- L146 "with a maximum of 700 m and a minimum of 100 m"

    Changed as suggested

- L194 "Despite low albedo (..) and high net radiation". What do you mean by despite?

    Here we were suggesting that although there is high absorption of incoming radiation the high rate of evapotranspiration results in cooler land surface temperatures. In order to avoid confusion we have changed this sentence and it now reads "Evapotranspiration in the Amazon has been shown to result in a net cooling effect (Bonan, 2008) therefore, areas with lower LST will be observed in areas of higher ET (Eltahir and Bras, 1994)".
    .

- L200 "as EVI is less sensitive"

    Changed as suggested

- L217 "due to our large sample size, we still have enough"

    Changed as suggested

- L220 / L232: both start with finally

    Changed: "Finally" was removed from the paragraph starting on L220

- L231 "SI.9.4"

    Changed as suggested

- L234 "which was then applied to an accumulation to calculate" This sentence is unclear to me.

    Changed: We opted to remove the reference to how seasonality is calculated as this is addressed in detail later in the paper. The sentence now reads "which was then used to calculate the seasonality of rainfall"

- L247 "the model overestimated deep WTD" / L458 "may be underestimated"

    Changed as suggested

- L251 "these depths were selected as they represent rooting depth"

    Changed as suggested

- L270 "crop vegetation has a low"

    Changed as suggested

- L310 "The WST and DST periods were selected .."

    Changed as suggested

- L318 "shallow rooted vegetation"

Changed as suggested

- L376 "effect of ground water depth on Land Surface Temperature"

  Changed as suggested

- L401 "between deep and shallow WTD"

  Changed as suggested

- L412 with between, do you mean among?

  Changed as suggested to Among

- L425 "we expect to see the largest influence of WTD in crop vegetation".

  Changed as suggested

- L428 "was not different between crop areas in .."

  Changed as suggested

- L430 replace "a high / a low" with for example a maximum / a minimum"

  Changed as suggested

- L440 "and therefore evaporative cooling could explain the cooler"

  Changed as suggested

- L476 "higher in shallow than deep WTD areas"

  Changed as suggested

- L475 "during the WST EVI is increasing / increases faster"

  Changed as suggested

- L531 "can vary between September at October". Do you mean September and October?

  Yes indeed Changed as suggested

- L561 the sentence "on a local scale show signs that .." is unclear to me.

  Changed: "So although our results are not directly relevant at regional or continental scales on a local scale shallow WTD areas are more productive than deep WTD."

Additional changed made to the manuscript

L284 & L286: "At first" and "Secondly" were removed from the paragraph